# CalFAT: Calibrated Federated Adversarial Training with Label Skewness

**Chen Chen**[*]
Zhejiang University

**Yuchen Liu**
Zhejiang University

**Xingjun Ma**
Fudan University

**Lingjuan Lyu**[†]
Sony AI

## Abstract

Recent studies have shown that, like traditional machine learning, federated learning (FL) is also vulnerable to adversarial attacks. To improve the adversarial robustness of FL, federated adversarial training (FAT) methods have been proposed to apply adversarial training locally before global aggregation. Although these methods demonstrate promising results on independent identically distributed (IID) data, they suffer from training instability on non-IID data with label skewness, resulting in degraded natural accuracy. This tends to hinder the application of FAT in real-world applications where the label distribution across the clients is often skewed. In this paper, we study the problem of FAT under label skewness, and reveal one root cause of the training instability and natural accuracy degradation issues: skewed labels lead to non-identical class probabilities and heterogeneous local models. We then propose a Calibrated FAT (CalFAT) approach to tackle the instability issue by calibrating the logits adaptively to balance the classes. We show both theoretically and empirically that the optimization of CalFAT leads to homogeneous local models across the clients and better convergence points.

## 1 Introduction

Federated learning (FL) is a privacy-aware learning paradigm that allows multiple participants (clients) to collaboratively train a global model without sharing their private data [23, 28, 43, 29]. In FL, each client follows the conventional machine learning procedure to train a local model on its own data and periodically uploads the local model updates to a central server for global aggregation. However, recent studies have shown that, like conventional machine learning, FL is also vulnerable to well-crafted adversarial examples [20, 47, 9, 45], i.e., at inference time, attackers can add small, human-perceptible adversarial perturbations to the test examples to fool the global model with high success rates. This raises security and reliability concerns on the implementation of FL in real-world scenarios where such a vulnerability could cause heavy losses [38]. For example, for cross-silo FL in the biomedical domain, a vulnerable global model may cause misdiagnosis, wrong medical treatments, or even the loss of lives. Similarly, in financial-based cross-silo FL, the lack of adversarial robustness may lead to huge financial losses. It is thus imperative to develop a robust FL method that can train adversarially robust global models resistant to different types of adversarial attacks.

In conventional machine learning, adversarial training (AT) has been shown to be one of the most effective defenses against adversarial attacks [22, 40, 4]. Since the local training in FL is the same as conventional machine learning, recent works [47, 9, 45] proposed to perform local AT to

---

[*]Work done during internship at Sony AI.
[†]Corresponding author.

improve the adversarial robustness of the global model. These methods in general are known as Federated Adversarial Training (FAT). AT has been found to be *more* challenging than standard training [3, 44, 42, 41, 4], as it generally requires more training data and larger-capacity models. Moreover, adversarial robustness may even be at odds with accuracy [30], meaning that the increase of robustness may inevitably decrease the natural accuracy (i.e., accuracy on natural test data). As a result, the natural accuracy of AT is much lower than standard training [5]. This phenomenon also exists in FL, i.e., FAT exhibits slower convergence and lower natural accuracy than standard FL, as mentioned by recent studies [47, 9].

Arguably, FAT will become more challenging if the data are non-independent and identically distributed (non-IID) across the clients. One typical non-IID setting that commonly exists in real-world applications is skewed label distribution [17], where different clients have different label distributions. In this paper, we study the problem of FAT on non-IID data with a particular focus on the challenging skewed label distribution setting (formally defined in Section 3.1). Under conventional training, Xu et al. [37] have shown that adversarially trained models introduce severe performance disparity across different classes. And such a disparity will be exacerbated under label skewness, ending up with much worse performance on the minority classes [33].

By far, only a few works have studied non-IID FAT in the current literature. Zizzo et al. [47] propose to perform AT on only part of the local data for better convergence, while standard training is applied to the rest of the local data. We term this method as MixFAT. Another relevant work called FedRBN [9] tackles a different problem: how to propagate federated robustness to low-resource clients. Although MixFAT and FedRBN have demonstrated promising results, they suffer from training instability and low natural accuracy issues when compared to standard FL, as we show in Figure 1. We also compare with the other four FAT baselines adapted from existing AT methods to FL, i.e., FedPGD, Fed-TRADES, FedMART, and FedGAIRAT. Unfortunately, these methods also exhibit slow convergence and much degraded final accuracy (details

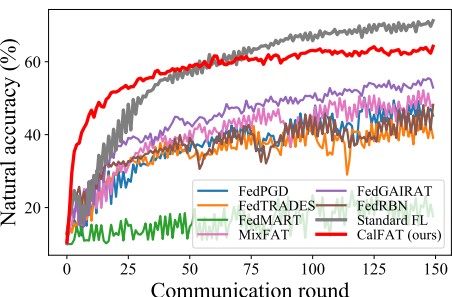

Figure 1: Natural accuracy and convergence of standard FL, our CalFAT, and 6 FAT baselines (FedPGD, FedTRADES, FedMART, MixFAT [47], FedGAIRAT and FedRBN [9]) under skewed label distribution with $\beta = 0.1$ (see Section 4).

can be found in Section 4.1). This motivates us to propose a novel method called *Calibrated Federated Adversarial Training* (CalFAT) for effective FAT on non-IID data with skewed label distribution. CalFAT tackles the training instability issue by calibrating the logits to give higher scores to the minority classes.

In summary, our main contributions are:

- *New insight:* We study the problem of FAT on non-IID data with skewed label distribution, and reveal one root cause of the training instability and natural accuracy degradation: skewed labels lead to non-identical class probabilities and heterogeneous local models.

- *Novel method:* We propose a novel method called CalFAT for FAT with label skewness, and show that the optimization of CalFAT can lead to homogeneous local models, and consequently, stable training, faster convergence, and better final performance.

- *High effectiveness*: Extensive experiments on 4 benchmark vision datasets across various settings prove the effectiveness of our CalFAT and its superiority over existing FAT methods.

## 2 Notation and Preliminaries

### 2.1 Notation

Suppose there are $m$ clients in FL with $i$ denoting the $i$-th client, e.g., $\mathcal{D}_i$ denotes the local data of client $i$ and $\theta_i$ denotes the parameters of its local model. We use $\hat{\theta}$ to denote the parameters of the global model. Subscript $j$ is the sample index, e.g., $(x_{ij}, y_{ij})$ denotes the $j$-th sample of client $i$ and its corresponding label with $y_{ij} \in \{1, \cdots, C\}$. Let $f_\theta(\cdot)$ be the local model $f(\cdot)$ (before softmax)

with parameter $\theta$. Superscript $l$ is the class index, e.g., $f^l(\cdot)$ denotes the logit output for class $l$. We denote the adversarial example of clean sample $x$ by $\widetilde{x}$. $[m]$ denotes the integer set $\{1, \cdots, m\}$. $p_i(x, y)$ denotes the joint distribution of input $x$ and label $y$ at client $i$, and accordingly, $p_i(y)$ is the marginal distribution of label $y$, $p_i(y \mid x)$ is the conditional distribution of label $y$ given input $x$ and $p_i(x \mid y)$ is the conditional distribution of input $x$ given label $y$.

## 2.2 Centralized Adversarial Training

Let $\mathcal{D} = \{x_j, y_j\}_{j=1}^n$ be the training dataset with $n$ samples. The cross-entropy loss $\ell_{ce}(f_\theta(x), y)$ for an input-label pair $(x, y)$ is defined as $\ell_{ce}(f_\theta(x), y) = -\log \sigma^y(f_\theta(x))$, where $\sigma^y(f) = \exp(f^y)/\sum_{l=1}^C \exp(f^l)$ is the softmax function, $C$ is the number of classes, and $f^l$ is the model output for class $l$. The objective function of the centralized adversarial training (AT) [22] can then be defined as $\min_\theta \sum_{j=1}^n \ell_{ce}(f_\theta(\widetilde{x}_j), y_j)/n$, where the adversarial example $\widetilde{x}_j$ can be generated by

$$\widetilde{x}_j = \arg\max_{x'_j \in \mathcal{B}_\epsilon(x_j)} \ell_{ce}(f_\theta(x'_j), y_j), \tag{1}$$

where $\mathcal{B}_\epsilon(x_j) = \{x' \mid \|x' - x_j\|_\infty < \epsilon\}$ is the closed ball of radius $\epsilon > 0$ centered at $x_j$, $\|\cdot\|_\infty$ is the $L_\infty$ norm, and $\widetilde{x}_j$ is the most adversarial sample within the $\epsilon$-ball.

A standard centralized AT method uses Projected Gradient Decent (PGD) to generate adversarial examples [22]. In particular, PGD iteratively generates an adversarial example $\widetilde{x}_j$ as follows:

$$x_j^{(k+1)} = \Pi_{\mathcal{B}_\epsilon(x_j^{(0)})}\left(x_j^{(k)} + \alpha \operatorname{sign}(\nabla_x \ell_{ce}(f_\theta(x_j^{(k)}), y_j))\right), \quad k = 0, \cdots, K-1, \tag{2}$$

where $k$ is the step number, $K$ is the total number of steps (i.e., $\widetilde{x}_j = x_j^{(K)}$), $\alpha > 0$ is the step size, $x_j^{(0)}$ is the natural sample, $x_j^{(k)}$ is the adversarial example generated at step $k$, $\Pi_{\mathcal{B}_\epsilon(x_j^{(0)})}$ is the projection function that projects the adversarial data onto the $\epsilon$-ball centered at $x_j^{(0)}$, and $\operatorname{sign}(\cdot)$ is the sign function.

By optimizing the model parameters on adversarial examples generated by PGD, centralized AT is able to train a model that is robust against adversarial attacks.

## 2.3 Federated Adversarial Training

The concept of federated adversarial training (FAT) was first introduced in [47] (i.e., MixFAT) to deal with the adversarial vulnerability of FL. MixFAT applies AT locally to improve the robustness of the global model. Suppose there are $m$ clients and each client $i$ has its local data $\mathcal{D}_i = \{x_{ij}, y_{ij}\}_{j=1}^{n_i}$ sampled from distribution $p_i(x, y)$ with $n_i = |\mathcal{D}_i|$ being the size of the local data. In MixFAT, each client $i$ optimizes its local model by minimizing the following objective:

$$\min_{\theta_i} \frac{1}{n_i}\left(\sum_{j=1}^{n'_i} \ell_{ce}(f_{\theta_i}(\widetilde{x}_{ij}), y_{ij}) + \sum_{j=n'_i+1}^{n_i} \ell_{ce}(f_{\theta_i}(x_{ij}), y_{ij})\right), \tag{3}$$

where $\widetilde{x}_{ij}$ is the PGD adversarial example of $x_{ij}$, $n'_i$ is a hyperparameter that controls the ratio of data for AT, and $\theta_i$ are the local model parameters. After training the local model for certain epochs, client $i$ uploads its local model parameters $\theta_i$ to the central server for aggregation. Note that MixFAT only applies AT to a proportion of the local data, mainly for convergence and stability considerations.

# 3 Calibrated Federated Adversarial Training (CalFAT)

## 3.1 Skewed Label Distribution Leads to Non-identical Class Probabilities

In this paper, we focus on one representative non-IID setting: *skewed label distribution* [19, 10], which is defined as follows.

**Definition 1** (Skewed label distribution). The label distribution across the clients is skewed, if for all $i \neq u$ and $i, u \in [m]$:

(a) there exists $y \in [C]$ such that $p_i(y) \neq p_u(y)$ and (b) $p_i(x \mid y) = p_u(x \mid y)$ for all $x, y$.

Condition (b) is to assume that, given a class $y$, $x$ is sampled with equal probability at different clients. Note that there exist different types of non-IID: label skew, non-identical class conditional, quantity skew, to name a few (Appendix K in [10]). The class conditional is often assumed to be identical (i.e., condition (b)) when studying the label skewness problem, which is the main focus of this work. When condition (b) does not hold, it becomes the non-identical class conditional problem.

**Lemma 1** (Non-identical class probabilities). *If the label distribution across the clients is skewed and the class conditionals have the same support, then the class probabilities $\{p_i(y \mid x) \mid i \in [m]\}$ are non-identical, i.e., for all $i \neq u$ and $i, u \in [m]$, there exist $x$, $y$ such that $p_i(y \mid x) \neq p_u(y \mid x)$.*

Lemma 1 implies that skewed label distribution gives rise to non-identical class probabilities $\{p_i(y \mid x) \mid i \in [m]\}$. The proof of Lemma 1 is given in Appendix A.

### 3.2 Standard Cross-entropy Leads to Heterogeneity

From a statistical point of view, each client $i$ in previous FAT methods estimates its local class probability $p_i(y \mid x)$ during local training [7]. More specifically, they assume that $p_i(y \mid x)$ can be parameterized by $\theta_i^*$ as:

$$p_i(y \mid x) = \hat{p}(y \mid x; \theta_i^*) = \sigma^y(f_{\theta_i^*}(x)), \tag{4}$$

where $\theta_i^*$ is the ground-truth parameters of the local class probability $p_i(y \mid x)$. According to Lemma 1, the class probabilities $\{p_i(y \mid x)\}$ are non-identical when there is a skewed label distribution. Therefore, the ground-truth parameters $\{\theta_i^* \mid i \in [m]\}$ are heterogeneous. We use the sample variance of the ground-truth parameters to measure such heterogeneity as follows:

$$(s^*)^2 = V(\theta_1^*, \ldots, \theta_m^*) = \frac{1}{m-1} \sum_{i=1}^{m} \left\| \theta_i^* - \frac{1}{m} \sum_{j=1}^{m} \theta_j^* \right\|^2. \tag{5}$$

Each client $i$ updates its local model parameters $\theta_i$ by optimizing the standard cross-entropy (CE) loss. The updated $\theta_i$ is the maximum likelihood estimate [1] of the ground-truth parameter $\theta_i^*$ [7]. Similarly, we use the sample variance [1] of the local model parameters to measure the heterogeneity of the local models:

$$s^2 = V(\theta_1, \ldots, \theta_m). \tag{6}$$

Larger sample variance implies higher model heterogeneity.

The following proposition suggests that the heterogeneity of local models originates from the heterogeneity of the local class probabilities.

**Proposition 1** (Heterogeneous local models). *Assume the label distribution across the clients is skewed. Let $\theta_i$ be the maximum likelihood estimate of $\theta_i^*$ in Eq. (4) given local data at client $i$. Then $s^2$ converges almost surely to a nonzero constant:*

$$s^2 \xrightarrow{a.s.} (s^*)^2 \neq 0, \tag{7}$$

*where $\xrightarrow{a.s.}$ represents the almost sure convergence.*

The proof of Proposition 1 is provided in Appendix B. $(s^*)^2$ measures the heterogeneity of the ground-truth parameters $\{\theta_i^* \mid i \in [m]\}$, which reflects the class probability difference across the clients as shown in Eq. (4).

Proposition 1 implies that the local models in previous FAT methods are heterogeneous when the label distribution across the clients is skewed. Since the local models are heterogeneous, aggregating these models tends to hurt the convergence and cause the divergence of the global model [17]. As shown in Figure 1, the training process of existing FAT methods is unstable and has much lower natural accuracy than the standard FL.

### 3.3 Learning Homogeneous Local Models by Calibration

Motivated by [24], we propose to re-parameterize the class probabilities. According to Bayes' formula [12],

$$p_i(y \mid x) = \frac{p_i(x \mid y) p_i(y)}{\sum_{l=1}^{C} p_i(x \mid l) p_i(l)}. \tag{8}$$

---

**Algorithm 1** Local training of CalFAT

---

**Input:**

Client $i$, global model parameters $\hat{\theta}$, local dataset $\mathcal{D}_i$, local epoch number $E$, and positive constant $\delta$

 1: **procedure** CLIENTUPDATE
 2:     $\theta_i \leftarrow \hat{\theta}$
 3:     Compute $\pi_i$ with $\mathcal{D}_i$ by $\pi_i^y = n_i^y/n_i + \delta, y \in [C]$
 4:     **for** local epoch=$1, \cdots, E$ **do**
 5:         **for** $j = 1, \cdots, n_i$ **do**
 6:             Sample $(x_{ij}, y_{ij})$ from $\mathcal{D}_i$
 7:             Generate adversarial example $\widetilde{x}_{ij} = \arg\max_{x'_{ij} \in \mathcal{B}_\epsilon(x_{ij})} \ell_{ckl}(f_{\theta_i}(x'_{ij}), f_{\theta_i}(x_{ij}), \pi_i)$
 8:         **end for**
 9:         $\theta_i \leftarrow \theta_i - \eta \frac{1}{n_i} \sum_{j=1}^{n_i} \nabla_{\theta_i} \ell_{cce}(f_{\theta_i}(\widetilde{x}_{ij}), y_{ij}, \pi_i)$
10:     **end for**
11:     **return** $\theta_i$
12: **end procedure**

---

On the right-hand side of the above equation: (1) the class priors can be easily computed by the relative frequencies [1]; and (2) more importantly, the class conditionals $\{p_i(x \mid y) \mid i \in [m]\}$ are *identical* across different clients (see Definition 1).

Inspired by the above observation, we propose an alternative parameterization of $p_i(y \mid x)$. Assume that for all $i \in [m]$, the class conditional $p_i(x \mid y)$ can be parameterized by $\theta^*$ as $p_i(x \mid y) = \hat{q}(x \mid y; \theta^*)$, where $\hat{q}(x \mid y; \theta^*)$ can be an arbitrary conditional probability function. Then, $p_i(y \mid x)$ can be re-parameterized by $\theta^*$ as follows:

$$p_i(y \mid x) = \hat{q}_i(y \mid x; \theta^*) = \frac{\hat{q}(x \mid y; \theta^*) \pi_i^y}{\sum_{l=1}^{C} \hat{q}(x \mid l; \theta^*) \pi_i^l}. \tag{9}$$

where

$$\pi_i^y = n_i^y/n_i + \delta, y \in [C]. \tag{10}$$

Here $\pi_i^y$ approximates the class prior $p_i(y)$, $n_i^y$ is the sample size of class $y$ on client $i$ and $\delta > 0$ is a small constant added for numerical stability purpose. During local updates, client $i$ uses its local data to update $\theta_i$, which makes $\theta_i$ the maximum likelihood estimate of $\theta^*$. The entire training procedure of our method is described in Section 3.4.

The following proposition suggests that the local models are homogeneous when trained with the above re-parameterization. The proof of Proposition 2 is provided in Appendix C.

**Proposition 2** (Homogeneous local models). *Assume the label distribution across the clients is skewed. Let $\theta_i$ be the maximum likelihood estimate of $\theta^*$ in Eq. (9) given local data at client $i$. Then $s^2$ converges almost surely to zero:*

$$s^2 \xrightarrow{a.s.} 0. \tag{11}$$

## 3.4   Details of CalFAT

The local training procedure of our proposed CalFAT is described in Algorithm 1. Specifically, we define $\hat{q}(x \mid y; \theta^*) = \exp(f_{\theta^*}^y(x))$. Then, we maximize the likelihood of $\hat{q}_i(y \mid x; \theta^*)$ for each client $i$, which is equivalent to minimizing the following objective:

$$\min_{\theta_i} \frac{1}{n_i} \sum_{j=1}^{n_i} \ell_{cce}(f_{\theta_i}(\widetilde{x}_{ij}), y_{ij}, \pi_i), \tag{12}$$

where $\ell_{cce}(\cdot, \cdot, \cdot)$ is the calibrated cross-entropy (CCE) loss and $\widetilde{x}_{ij}$ is the adversarial example of $x_{ij}$. The CCE loss is defined as:

$$\ell_{cce}(f_{\theta_i}(\widetilde{x}_{ij}), y_{ij}, \pi_i) = -\log \sigma^{y_{ij}}(f_{\theta_i}(\widetilde{x}_{ij}) + \log \pi_i). \tag{13}$$

As discussed in Section 3.3, minimizing the above CCE loss mitigates the heterogeneity of the local models, which can lead to improved convergence and performance of the global model.

Table 1: Natural and robust accuracy (%) on different datasets. The best results are in **bold**.

| Dataset | CIFAR10 | | | | | | CIFAR100 | | | | | |
|---|---|---|---|---|---|---|---|---|---|---|---|---|
| Metric | Natural | FGSM | BIM | CW | PGD-20 | AA | Natural | FGSM | BIM | CW | PGD-20 | AA |
| MixFAT | $53.35 \pm 0.11$ | $29.14 \pm 0.10$ | $26.31 \pm 0.17$ | $22.79 \pm 0.12$ | $26.27 \pm 0.11$ | $21.89 \pm 0.13$ | $34.43 \pm 0.13$ | $15.69 \pm 0.13$ | $14.60 \pm 0.14$ | $11.31 \pm 0.17$ | $14.36 \pm 0.20$ | $9.06 \pm 0.11$ |
| FedPGD | $46.96 \pm 0.16$ | $28.70 \pm 0.19$ | $26.59 \pm 0.18$ | $24.38 \pm 0.17$ | $26.74 \pm 0.18$ | $22.47 \pm 0.11$ | $33.96 \pm 0.14$ | $16.07 \pm 0.08$ | $14.68 \pm 0.10$ | $11.67 \pm 0.10$ | $14.67 \pm 0.15$ | $10.87 \pm 0.12$ |
| FedTRADES | $46.06 \pm 0.12$ | $27.75 \pm 0.17$ | $26.32 \pm 0.09$ | $22.86 \pm 0.10$ | $26.31 \pm 0.12$ | $21.70 \pm 0.09$ | $29.55 \pm 0.10$ | $15.01 \pm 0.06$ | $14.11 \pm 0.11$ | $10.58 \pm 0.03$ | $14.30 \pm 0.13$ | $9.53 \pm 0.09$ |
| FedMART | $25.67 \pm 0.21$ | $18.50 \pm 0.18$ | $18.21 \pm 0.22$ | $15.22 \pm 0.17$ | $18.10 \pm 0.22$ | $14.41 \pm 0.20$ | $19.96 \pm 0.17$ | $13.00 \pm 0.19$ | $12.91 \pm 0.14$ | $9.92 \pm 0.21$ | $12.83 \pm 0.18$ | $8.57 \pm 0.14$ |
| FedGAIRAT | $48.42 \pm 0.08$ | $29.30 \pm 0.09$ | $26.55 \pm 0.07$ | $22.78 \pm 0.12$ | $27.20 \pm 0.08$ | $21.96 \pm 0.07$ | $34.92 \pm 0.05$ | $16.18 \pm 0.06$ | $15.37 \pm 0.10$ | $11.80 \pm 0.05$ | $14.90 \pm 0.03$ | $9.41 \pm 0.05$ |
| FedRBN | $47.80 \pm 0.06$ | $26.87 \pm 0.07$ | $26.25 \pm 0.03$ | $22.00 \pm 0.01$ | $26.30 \pm 0.09$ | $21.33 \pm 0.09$ | $28.55 \pm 0.07$ | $14.69 \pm 0.04$ | $13.41 \pm 0.08$ | $9.71 \pm 0.08$ | $14.15 \pm 0.12$ | $8.83 \pm 0.08$ |
| CalFAT (ours) | $\mathbf{64.69 \pm 0.08}$ | $\mathbf{35.03 \pm 0.12}$ | $\mathbf{31.50 \pm 0.07}$ | $\mathbf{24.69 \pm 0.11}$ | $\mathbf{31.12 \pm 0.11}$ | $\mathbf{22.91 \pm 0.08}$ | $\mathbf{44.57 \pm 0.10}$ | $\mathbf{17.63 \pm 0.10}$ | $\mathbf{15.60 \pm 0.11}$ | $\mathbf{12.01 \pm 0.11}$ | $\mathbf{15.21 \pm 0.07}$ | $\mathbf{11.49 \pm 0.08}$ |

| Dataset | SVHN | | | | | | ImageNet subset | | | | | |
|---|---|---|---|---|---|---|---|---|---|---|---|---|
| Metric | Natural | FGSM | BIM | CW | PGD-20 | AA | Natural | FGSM | BIM | CW | PGD-20 | AA |
| MixFAT | $19.57 \pm 0.10$ | $19.61 \pm 0.12$ | $19.66 \pm 0.12$ | $19.66 \pm 0.11$ | $19.75 \pm 0.11$ | $14.80 \pm 0.07$ | $33.53 \pm 0.06$ | $19.47 \pm 0.02$ | $18.48 \pm 0.10$ | $16.15 \pm 0.07$ | $18.39 \pm 0.02$ | $11.98 \pm 0.06$ |
| FedPGD | $19.55 \pm 0.08$ | $19.33 \pm 0.09$ | $19.37 \pm 0.08$ | $19.68 \pm 0.05$ | $19.52 \pm 0.09$ | $13.64 \pm 0.10$ | $30.87 \pm 0.12$ | $18.88 \pm 0.13$ | $17.95 \pm 0.10$ | $16.07 \pm 0.11$ | $18.40 \pm 0.16$ | $11.34 \pm 0.08$ |
| FedTRADES | $56.96 \pm 0.13$ | $36.92 \pm 0.13$ | $35.15 \pm 0.05$ | $31.08 \pm 0.14$ | $34.90 \pm 0.15$ | $30.37 \pm 0.11$ | $30.22 \pm 0.14$ | $18.67 \pm 0.13$ | $17.99 \pm 0.21$ | $16.23 \pm 0.13$ | $17.82 \pm 0.12$ | $11.81 \pm 0.10$ |
| FedMART | $19.85 \pm 0.16$ | $19.94 \pm 0.16$ | $19.71 \pm 0.16$ | $19.85 \pm 0.15$ | $19.79 \pm 0.17$ | $14.64 \pm 0.14$ | $26.47 \pm 0.18$ | $16.40 \pm 0.18$ | $15.53 \pm 0.17$ | $14.43 \pm 0.13$ | $15.40 \pm 0.21$ | $9.34 \pm 0.14$ |
| FedGAIRAT | $58.41 \pm 0.11$ | $38.30 \pm 0.12$ | $36.52 \pm 0.09$ | $31.24 \pm 0.15$ | $36.69 \pm 0.13$ | $31.63 \pm 0.06$ | $34.25 \pm 0.12$ | $19.62 \pm 0.10$ | $19.28 \pm 0.14$ | $16.78 \pm 0.14$ | $19.18 \pm 0.12$ | $11.80 \pm 0.09$ |
| FedRBN | $53.88 \pm 0.04$ | $34.48 \pm 0.08$ | $32.52 \pm 0.02$ | $27.99 \pm 0.02$ | $32.32 \pm 0.03$ | $28.35 \pm 0.05$ | $29.35 \pm 0.09$ | $18.76 \pm 0.03$ | $17.25 \pm 0.12$ | $15.07 \pm 0.10$ | $18.05 \pm 0.09$ | $11.42 \pm 0.12$ |
| CalFAT (ours) | $\mathbf{84.15 \pm 0.07}$ | $\mathbf{48.38 \pm 0.11}$ | $\mathbf{42.04 \pm 0.07}$ | $\mathbf{31.66 \pm 0.04}$ | $\mathbf{41.68 \pm 0.11}$ | $\mathbf{32.57 \pm 0.10}$ | $\mathbf{49.89 \pm 0.11}$ | $\mathbf{22.31 \pm 0.17}$ | $\mathbf{19.99 \pm 0.09}$ | $\mathbf{17.42 \pm 0.12}$ | $\mathbf{19.97 \pm 0.14}$ | $\mathbf{12.30 \pm 0.07}$ |

In previous FAT methods, heterogeneous local models tend to give higher scores to the majority classes while lower scores to the minority classes. By contrast, our CalFAT encourages local models to give higher scores to the minority classes by adding a class-wise prior $\log \pi_i^l$ to the logits. Also different from MixFAT that trains the local models on both natural and adversarial data, our CalFAT trains the local models only on adversarial examples. Extensive empirical experiments are conducted in Section 4.1 to show the impact of using only adversarial data for optimization.

**Adversarial example generation.** Inspired by [40], we generate the adversarial examples by maximizing the following calibrated Kullback–Leibler (CKL) divergence loss:

$$\widetilde{x}_{ij} = \underset{x'_{ij} \in \mathcal{B}_\epsilon(x_{ij})}{\arg\max} \ell_{ckl}(f_{\theta_i}(x'_{ij}), f_{\theta_i}(x_{ij}), \pi_i), \tag{14}$$

where $\ell_{ckl}(\cdot, \cdot, \cdot)$ is the CKL loss defined as:

$$\ell_{ckl}(f_{\theta_i}(x'_{ij}), f_{\theta_i}(x_{ij}), \pi_i) = -\sum_{y=1}^{C} \sigma^y(f_{\theta_i}(x_{ij}) + \log \pi_i) \log \sigma^y(f_{\theta_i}(x'_{ij}) + \log \pi_i), \tag{15}$$

where $\log \pi_i$ is the same as in our CCE loss. Following centralized AT [22], we also use PGD to solve Eq. (14).

After training the local model for certain epochs following the above procedure, each client $i$ uploads the model parameters $\theta_i$ to the server for aggregation. To be consistent with the most recent FAT methods [47, 9], we adopt the most widely used FedAvg [23] as the default aggregation framework. Our method is compatible with other FL frameworks (e.g., FedProx [16] and Scaffold [11]), as we will show in Section 4.1.

## 4 Experiments

**Data configurations.** Our experiments are conducted on 4 real-world datasets: CIFAR10 [13], CIFAR100 [13], SVHN [25], and ImageNet subset [6]. To simulate label skewness, we sample $p_i^l \sim Dir(\beta)$ and allocate a $p_i^l$ proportion of the data of label $l$ to client $i$, where $Dir(\beta)$ is the Dirichlet distribution with a concentration parameter $\beta$ [39]. By default, we set $\beta = 0.1$ to simulate a highly skewed label distribution that widely exists in reality.

**Baselines.** We compare our proposed CalFAT with two state-of-the-art FAT methods: MixFAT [47] and FedRBN [9]. We also investigate the combination of the state-of-the-art centralized AT methods with FL, i.e., we apply standard PGD [22], TRADES [40], MART [34]), and GAIRAT [44] to FL, and term them as FedPGD, FedTRADES, FedMART, and FedGAIRAT, respectively.

**Evaluation metrics.** We report the natural test accuracy (Natural) and robust test accuracy under the most representative attacks, i.e., FGSM [35], BIM [15], PGD-20 [22], CW [2], and AA [5]. We run the experiment for 5 times and report the mean and standard deviation. More detailed experimental setup is provided in Appendix D.1.

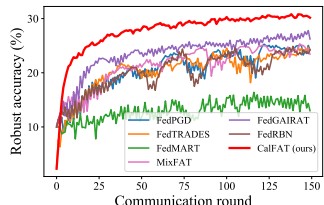
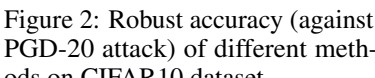
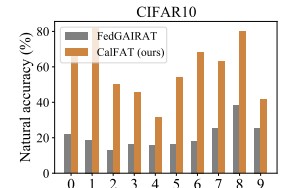
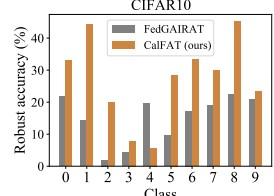

Figure 2: Robust accuracy (against PGD-20 attack) of different methods on CIFAR10 dataset.

Figure 3: Per-class natural accuracy and robust accuracy (against PGD-20 attack) of CalFAT and the best baseline (FedGAIRAT) on CIFAR10 dataset.

## 4.1 Main Results

**Evaluation on different datasets.** Table 1 shows the results of all methods on CIFAR10, CIFAR100, SVHN, and ImageNet subset. From the table, we can observe that:

(1) Our CalFAT achieves the best robustness on all datasets, validating the efficacy of our CalFAT. For example, CalFAT outperforms the best baseline method (FedGAIRAT) by 10.20% on SVHN dataset under FGSM attack.

(2) Our CalFAT shows a significant improvement in natural accuracy compared to other baselines. For example, CalFAT can improve the natural accuracy of the best baseline method (FedGAIRAT) by 25.63% on SVHN dataset. We hypothesise that the reason lies in the homogeneity of local models in our CalFAT, which leads to better convergence and higher clean accuracy.

(3) All methods demonstrate the worst performance on CIFAR100 and ImageNet subset datasets. We conjecture that this is because there are more classes in these two datasets, making federated training substantially harder. Nevertheless, our CalFAT still achieves the best performance.

**Learning curves of different methods.** To visually compare our CalFAT with all the baselines, we plot the learning curves (i.e., performance across different communication rounds) of all methods in Figure 1 and Figure 2. As can be observed, CalFAT achieves the best natural accuracy and robust accuracy across almost the entire training process, which indicates that the design of our CalFAT is profitable for different federated learning stages.

Moreover, our CalFAT is fairly stable during the whole training process while the accuracy curves of other baselines oscillate strongly. Such oscillations lead to bad convergence and low performance. We hypothesize that the heterogeneity of local models in the baseline methods is the main cause of the unstable training.

**Combining calibration loss with other FAT methods.** To further show the effectiveness of our calibration loss, we combine it with four FAT methods (MixFAT, FedPGD, FedTRADES, and FedMART) and name them MixFAT + Calibration, FedPGD + Calibration, FedTRADES + Calibration, and FedMART + Calibration, respectively. We compare these calibration loss-based FAT methods with their original versions in Table 2. It is evident that, by introducing our calibration loss into their objectives, all FAT methods can be improved. These results confirm the importance of class calibration for FAT with label skewness.

Table 2: Combining FAT methods with different losses.

| Metric | Natural | PGD-20 |
|---|---|---|
| MixFAT | $53.35 \pm 0.11$ | $26.27 \pm 0.11$ |
| MixFAT + LogitAdj | $57.53 \pm 0.21$ | $27.65 \pm 0.16$ |
| MixFAT + RoBal | $58.25 \pm 0.13$ | $27.86 \pm 0.10$ |
| MixFAT + Calibration (ours) | $\mathbf{60.23} \pm 0.19$ | $\mathbf{28.67} \pm 0.14$ |
| FedPGD | $46.96 \pm 0.16$ | $26.74 \pm 0.18$ |
| FedPGD + LogitAdj | $59.79 \pm 0.15$ | $28.84 \pm 0.12$ |
| FedPGD + RoBal | $61.48 \pm 0.07$ | $29.51 \pm 0.07$ |
| FedPGD + Calibration (ours) | $\mathbf{63.91} \pm 0.13$ | $\mathbf{30.72} \pm 0.16$ |
| FedTRADES | $46.06 \pm 0.12$ | $26.31 \pm 0.12$ |
| FedTRADES + LogitAdj | $58.26 \pm 0.20$ | $27.92 \pm 0.19$ |
| FedTRADES + RoBal | $59.25 \pm 0.23$ | $28.63 \pm 0.08$ |
| FedTRADES + Calibration (ours) | $\mathbf{63.12} \pm 0.10$ | $\mathbf{30.27} \pm 0.23$ |
| FedMART | $25.67 \pm 0.21$ | $18.10 \pm 0.22$ |
| FedMART + LogitAdj | $42.01 \pm 0.10$ | $24.92 \pm 0.02$ |
| FedMART + RoBal | $44.26 \pm 0.22$ | $25.57 \pm 0.17$ |
| FedMART + Calibration (ours) | $\mathbf{48.85} \pm 0.08$ | $\mathbf{27.19} \pm 0.11$ |
| CalFAT (ours) | $\mathbf{64.69} \pm 0.08$ | $\mathbf{31.12} \pm 0.11$ |

Table 3: Natural and robust accuracy (%) across different FL frameworks on CIFAR10 dataset.

| FL framework | FedProx | | | Scaffold | | |
|---|---|---|---|---|---|---|
| Metric | Natural | PGD-20 | AA | Natural | PGD-20 | AA |
| MixFAT | $53.75 \pm 0.16$ | $29.61 \pm 0.19$ | $21.59 \pm 0.27$ | $55.27 \pm 0.20$ | $28.78 \pm 0.15$ | $21.26 \pm 0.11$ |
| FedPGD | $49.57 \pm 0.18$ | $28.48 \pm 0.17$ | $21.31 \pm 0.18$ | $49.52 \pm 0.14$ | $27.46 \pm 0.21$ | $20.27 \pm 0.15$ |
| FedTRADES | $48.14 \pm 0.20$ | $27.75 \pm 0.17$ | $21.13 \pm 0.21$ | $47.78 \pm 0.23$ | $27.31 \pm 0.16$ | $20.04 \pm 0.16$ |
| FedMART | $28.32 \pm 0.22$ | $19.32 \pm 0.23$ | $15.91 \pm 0.25$ | $27.80 \pm 0.17$ | $20.03 \pm 0.26$ | $16.85 \pm 0.15$ |
| FedGAIRAT | $49.61 \pm 0.20$ | $29.34 \pm 0.11$ | $21.33 \pm 0.18$ | $49.54 \pm 0.21$ | $27.23 \pm 0.25$ | $20.16 \pm 0.09$ |
| FedRBN | $47.26 \pm 0.13$ | $26.63 \pm 0.15$ | $20.46 \pm 0.06$ | $49.77 \pm 0.09$ | $28.37 \pm 0.12$ | $20.32 \pm 0.06$ |
| CalFAT | $\mathbf{66.32} \pm 0.08$ | $\mathbf{32.79} \pm 0.13$ | $\mathbf{22.83} \pm 0.11$ | $\mathbf{67.16} \pm 0.06$ | $\mathbf{32.94} \pm 0.06$ | $\mathbf{21.94} \pm 0.05$ |

**Comparison with state-of-the-art long-tail learning methods.** We also compare our calibration loss with the losses used by long-tail learning methods LogitAdj [24] and RoBal [36]. In particular, we combine four FAT methods (MixFAT, FedPGD, FedTRADES, and FedMART) with the above three losses (LogitAdj loss, RoBal loss, and our calibration loss), and train the models following the same default setting. As shown in Table 2, both LogitAdj-based methods and RoBal-based methods have lower natural and robust accuracies than calibration loss-based methods. This indicates that our calibration loss is more suitable for FAT than other long-tail learning losses.

## 4.2 Performance on Different Classes

We further compare the per-class performance of our CalFAT with the best baseline FedGAIRAT. First, we use a well-trained model to initialize a global model. Second, the global model distributes the model parameter to all clients. Third, the local clients train their local models with their local data for 1 epoch. Then, we report the per-class average performance of all clients for each class. For fair comparison, we use the same well-trained model for initialization and the same data partition on each client for CalFAT and FedGAIRAT.

In Figure 3, we report the per-class natural and robust accuracies of CalFAT and FedGAIRAT on CIFAR10. As shown in these figures, the average performance of most classes of CalFAT is much higher than FedGAIRAT. We also report the per-class performance of each client on CIFAR10 in Appendix D.2. In FedGAIRAT, due to the highly skewed label distribution, the prediction of each client is highly biased to the majority classes, which leads to high performance on the majority classes and low performance (even 0% accuracy) on the minority classes. By contrast, in CalFAT, each client has higher performance on most classes. This verifies that the calibrated cross-entropy loss can indeed improve the performance on the minority classes, and further improve the overall performance of the model. Moreover, we report the per-class average performance on SVHN in Appendix D.3. Our CalFAT also outperforms the best baseline across most of the classes on SVHN.

## 4.3 Results on Different FL Frameworks and Network Architectures

**Evaluation on different FL frameworks.** Besides FedAvg [23], we also conduct experiments on other FL frameworks, i.e., FedProx [16] and Scaffold [11]. The results for all methods on FedProx and Scaffold are given in Table 3. It shows that our CalFAT exhibits better natural and robust accuracies than all baseline methods on all FL frameworks, which indicates the high comparability of our CalFAT with different aggregation algorithms.

**Evaluation on different network architectures.** We also compare CalFAT with baselines on different network architectures, i.e., CNN [23], VGG-8 [27], and ResNet-18 [8]. For CNN, we use the same architecture as [23]. VGG-8 and ResNet-18 are two widely used architectures in deep learning. The results on CIFAR10 dataset are shown in Appendix D.4. CalFAT outperforms all baselines, which further validates the superiority of CalFAT with different network architectures.

## 4.4 Feature Visualization

To better understand the efficacy of CalFAT, we visualize the learned features extracted from the second last layer of FedTRADES (the best baseline) and CalFAT trained on SVHN dataset in Appendix D.5. The features are projected into a 2-dimensional space via t-SNE [31]. It shows that

samples from different classes are mixed together in FedTRADES, indicating its low performance. For instance, Class 6 (pink) and Class 8 (khaki) are hard to separate in FedTRADES while these 2 classes can be well separated by CalFAT. This illustration verifies that the server cannot learn a global model with good inter-class separability if the local models are heterogeneous. By contrast, CalFAT can well separate different classes thus can achieve better overall performance.

### 4.5 Results under IID settings

Besides the non-IID setting, we also conduct an experiment under the IID setting. The results are shown in Appendix D.6 where it shows that our CalFAT achieves the best robustness (under PGD-20 attack). Compared to the non-IID setting, all FAT methods demonstrate much better performance under the IID setting. This indicates that existing FAT methods can easily handle IID data yet face substantial challenges when the data is non-IID.

### 4.6 Ablation Studies

**Impact of the number of clients.** To show the generality of CalFAT, we train CalFAT with different numbers of clients $m$. Table 7 in Appendix D.7 reports the results for $m = \{20, 50, 100\}$. As expected, CalFAT achieves the best performance across all $m$. As $m$ increases, the performance of all methods decreases. We conjecture that this is because more clients in FAT makes the training harder to converge. However, our CalFAT can still achieve 41.23% natural accuracy when there are 100 clients, outperforming other baselines by a large margin.

**Impact of skewed label distribution.** We observe that the performance of FAT defense is closely related to label skewness. We thus investigate the impact of skewed label distribution by varying the Dirichlet parameter $\beta = \{0.05, 0.2, 0.3\}$ and report the results on CIFAR10 in Table 8 in Appendix D.8. Not surprisingly, our CalFAT outperforms all baselines under all $\beta$'s. This further verifies the consistent effectiveness of CalFAT under different levels of label skewness.

Note that as $\beta$ decreases (i.e., the labels on each client are more imbalanced), the performance of all methods drop rapidly. For example, the natural accuracy of FedMART drops from 38.38% to 29.84% as $\beta$ decreases from 0.2 to 0.05. This indicates that all methods are hard to train a good model in extremely skewed label distribution scenarios. However, our CalFAT still achieves 61.00% natural accuracy and 32.40% robust accuracy (against FGSM attack) when $\beta = 0.05$, which are much higher than all the baselines.

**Contribution of the calibrated loss functions.** As shown in Eq. (13) and Eq. (15), for each client $i$, we have two new loss functions: a CCE loss $\ell_{cce}(\cdot, \cdot, \cdot)$ for optimization and a CKL loss $\ell_{ckl}(\cdot, \cdot, \cdot)$ for generating the adversarial examples. This naturally raises a question: how do these two loss functions contribute to CalFAT? To answer this question, we conduct leave-one-out tests by removing the CCE loss (w/o $\ell_{cce}(\cdot, \cdot, \cdot)$) or removing the CKL loss (w/o $\ell_{ckl}(\cdot, \cdot, \cdot)$) from the overall optimization objective. As illustrated in Appendix D.9, w/o $\ell_{cce}(\cdot, \cdot, \cdot)$ leads to poor performance, which implies that CCE loss plays an important role in CalFAT. Besides, if we only use the CCE loss (i.e., w/o $\ell_{ckl}(\cdot, \cdot, \cdot)$), we can obtain a much better performance, but it still underperforms CalFAT. All these results indicate that the CCE loss is the most important part of CalFAT, whilst the CKL loss can further increase the performance of CalFAT. The combination of both loss functions leads to the best performance.

**Impact of the ratio of adversarial data.** Here, we conduct experiments with different ratios of adversarial data used in CalFAT and report the robust accuracy (against PGD-20 attack) in Appendix D.10. Ratios $r=0$ and $r=1$ stand for training the model on only natural data and only adversarial data, respectively. Overall, $r=1$ produces the best robustness, meaning that training on only adversarial data can better enhance the adversarial robustness of our CalFAT.

## 5 Conclusion

In this paper, we studied the challenging problem of Federated Adversarial Training (FAT) with label skewness and proposed a novel Calibrated Federated Adversarial Training (CalFAT) to simultaneously achieve stable training, better convergence, and natural accuracy and robustness in FL. CalFAT

calibrates the model prediction and trains homogeneous local models across different clients by automatically assigning higher scores to the minority classes. Extensive experiments on multiple datasets under various settings validate the effectiveness of CalFAT. Our work can serve as a simple but strong baseline for accurate and robust FAT. For future work, we will continue to improve FAT under other non-IID settings such as feature skewness and quantity skewness [10, 46].

## Acknowledgement

This work is funded by Sony AI. This work is also supported by the National Key R&D Program of China (Grant No. 2021ZD0112804) and the National Natural Science Foundation of China (Grant No. 62276067).

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
