# A   Proof of Lemma 1

**Lemma 1** (Non-identical class probabilities). *If the label distribution across the clients is skewed and the class conditionals have the same support, then the class probabilities $\{p_i(y \mid x) \mid i \in [m]\}$ are non-identical, i.e., for all $i \neq u$ and $i, u \in [m]$, there exists $x$, $y$ such that $p_i(y \mid x) \neq p_u(y \mid x)$.*

*Proof. Case 1:* For all $y \in [C]$, $p_i(y), p_u(y) > 0$ or $p_i(y) = p_u(y) = 0$ .

We prove the result by contradiction. Assume that $p_i(y \mid x) = p_u(y \mid x)$ holds for all $x, y \in [C]$.

Consider $y \in [C]$ so that $p_i(y), p_u(y) > 0$. For all $x$, $p_i(x \mid y) > 0$

$$p_i(y \mid x) = p_u(y \mid x). \tag{16}$$

According to the Bayes' rule,

$$\frac{p_i(x \mid y)p_i(y)}{p_i(x)} = \frac{p_u(x \mid y)p_u(y)}{p_u(x)} \tag{17}$$

Cancel the $p_i(x \mid y) = p_u(x \mid y) \neq 0$ and obtain

$$\frac{p_i(y)}{p_i(x)} = \frac{p_u(y)}{p_u(x)}. \tag{18}$$

Take the reciprocal of both sides,

$$\frac{p_i(x)}{p_i(y)} = \frac{p_u(x)}{p_u(y)}. \tag{19}$$

Calculate the integral of both sides:

$$\int \frac{p_i(x)}{p_i(y)} dx = \int \frac{p_u(x)}{p_u(y)} dx, \tag{20}$$

$$\Rightarrow \frac{1}{p_i(y)} = \frac{1}{p_u(y)} \tag{21}$$

$$\Rightarrow p_i(y) = p_u(y). \tag{22}$$

This result contradicts the fact that there exists $y \in [C]$ such that $p_i(y) = p_u(y)$. Therefore, we conclude that the assumption must be false and that its opposite there exists $x, y \in [C]$ such that $p_i(y \mid x) \neq p_u(y \mid x)$ must be true in this case.

*Case 2:* There exists $y \in [C]$ that satisfies $p_i(y) > 0$, $p_u(y) = 0$ or $p_i(y) = 0$, $p_u(y) > 0$. Without loss of generality, we consider $p_i(y) > 0$ and $p_u(y) = 0$.

Take $x$ so that $p_i(x \mid y) > 0$, then according to Bayes' formula,

$$p_i(y \mid x) = \frac{p_i(x \mid y)p_i(y)}{p_i(x)} > 0, \tag{23}$$

$$p_u(y \mid x) = \frac{p_u(x \mid y)p_u(y)}{p_u(x)} = 0. \tag{24}$$

Therefore, $p_i(y \mid x) \neq p_i(y \mid x)$, which completes the proof.

$\square$

# B   Proof of Proposition 1

**Proposition 1** (Heterogeneous local models). *Assume the label distribution across the clients is skewed. Let $\theta_i$ be the maximum likelihood estimate of $\theta_i^*$ in Eq. (4) given local data at client $i$. Then $s^2$ converges almost surely to a nonzero constant:*

$$s^2 \xrightarrow{a.s.} (s^*)^2 \neq 0,$$

*where $\xrightarrow{a.s.}$ represents the almost sure convergence.*

*Proof.* According to the definition of sample variance, the convergence of local model parameters implies the convergence of $s^2$:

$$\left\{ \lim_{n_1,\ldots,n_m \to \infty} s^2 = (s^*)^2 \right\} \supseteq \left\{ \lim_{n_i \to \infty} \theta_i = \theta_i^*, \forall i \in [m] \right\}. \tag{25}$$

Then since probability is monotonic, we have

$$\Pr\left\{ \lim_{n_1,\ldots,n_m \to \infty} s^2 = (s^*)^2 \right\} \geq \Pr\left\{ \lim_{n_i \to \infty} \theta_i = \theta_i^*, \forall i \in [m] \right\}. \tag{26}$$

Since the sampling on different clients is independent, $\theta_i$ are independent, we have:

$$\Pr\left\{ \lim_{n_i \to \infty} \theta_i = \theta_i^*, \forall i \in [m] \right\} = \prod_{i=1}^{m} \Pr\left\{ \lim_{n_i \to \infty} \theta_i = \theta_i^* \right\}. \tag{27}$$

According to [32], the MLE $\theta_i$ is a consistent estimate of $\theta_i^*$:

$$\Pr\left\{ \lim_{n_i \to \infty} \theta_i = \theta_i^* \right\} = 1, \quad i \in [m]. \tag{28}$$

By combining Eq. (26), Eq. (27) and Eq. (28), it follows that

$$\Pr\left\{ \lim_{n_1,\ldots,n_m \to \infty} s^2 = (s^*)^2 \right\} \geq 1, \tag{29}$$

which implies

$$\Pr\left\{ \lim_{n_1,\ldots,n_m \to \infty} s^2 = (s^*)^2 \right\} = 1 \quad \Rightarrow \quad s^2 \xrightarrow{\text{a.s.}} (s^*)^2. \tag{30}$$

$\square$

## C  Proof of Proposition 2

**Proposition 2** (Homogeneous local models). *Assume the label distribution across the clients is skewed. Let $\theta_i$ be the maximum likelihood estimate of $\theta^*$ in Eq. (9) given local data at client $i$. Then $s^2$ converges almost surely to zero:*

$$s^2 \xrightarrow{\text{a.s.}} 0.$$

*Proof.* According to the definition of sample variance, the convergence of local model parameters implies the convergence of $s^2$:

$$\left\{ \lim_{n_1,\ldots,n_m \to \infty} s^2 = 0 \right\} \supseteq \left\{ \lim_{n_i \to \infty} \theta_i = \theta^*, \forall i \in [m] \right\}. \tag{31}$$

Then since probability is monotonic, we have

$$\Pr\left\{ \lim_{n_1,\ldots,n_m \to \infty} s^2 = 0 \right\} \geq \Pr\left\{ \lim_{n_i \to \infty} \theta_i = \theta^*, \forall i \in [m] \right\}. \tag{32}$$

Since the sampling on different clients is independent, $\theta_i$ are independent, we have:

$$\Pr\left\{ \lim_{n_i \to \infty} \theta_i = \theta^*, \forall i \in [m] \right\} = \prod_{i=1}^{m} \Pr\left\{ \lim_{n_i \to \infty} \theta_i = \theta^* \right\}. \tag{33}$$

According to [32], the MLE $\theta_i$ is a consistent estimate of $\theta^*$:

$$\Pr\left\{ \lim_{n_i \to \infty} \theta_i = \theta^* \right\} = 1, \quad i \in [m]. \tag{34}$$

By combining Eq. (32), Eq. (33) and Eq. (34), it follows that

$$\Pr\left\{ \lim_{n_1,\ldots,n_m \to \infty} s^2 = 0 \right\} \geq 1, \tag{35}$$

which implies

$$\Pr\left\{ \lim_{n_1,\ldots,n_m \to \infty} s^2 = 0 \right\} = 1 \quad \Rightarrow \quad s^2 \xrightarrow{\text{a.s.}} 0. \tag{36}$$

$\square$

# D Experimental Setup and Additional Experiments

## D.1 Detailed Experimental Setup

**Datasets.** Our experiments are conducted on 4 real-world datasets: CIFAR10 [13], CIFAR100 [13], SVHN [25], and ImageNet subset [6]. The ImageNet subset is generated according to [18], which consists of 12 labels. We resize the original image (with size 224*224*3) to 64*64*3 for fast training.

**Data partition.** To simulate real-world statistical heterogeneity, we use Dirichlet distribution to generate non-IID data across clients [39]. In particular, we sample $p_i^l \sim Dir(\beta)$ and allocate a $p_i^l$ proportion of the data of label $l$ to client $i$, where $Dir(\beta)$ is the Dirichlet distribution with a concentration parameter $\beta$. To simulate a highly skewed label distribution that widely exists in reality, we set $\beta = 0.1$ as default. We visualize the label distribution of 5 clients on CIFAR10 dataset (when $\beta = 0.1$) in Figure 4. The number in the figure stands for the number of training samples associated with the corresponding label in one particular client. As shown in the figure, the label distribution is highly skewed and each client has relatively few data (even no data) on some classes.

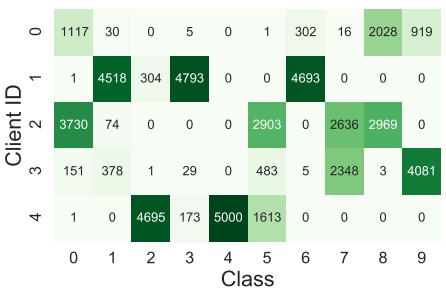

Figure 4: Label distribution of CIFAR10 among 5 different clients.

**Metric.** For evaluation, we report the natural test accuracy (Natural) on natural test data and the robust test accuracy on adversarial test data. The adversarial test data are generated by FGSM (fast gradient sign method) [35], BIM (basic iterative method with 20 steps) [15], PGD-20 (projected gradient descent with 20 steps) [22], CW (CW with 20 steps) [2], and AA (auto attack) [5] with the same perturbation bound $\epsilon = 8/255$. The step sizes for BIM, PGD-20 attack, and CW attack are $2/255$.

**Setting.** In our experiments, we consider $||\widetilde{x} - x||_\infty < \epsilon$ with the same $\epsilon$ for both training and evaluation. To generate the most adversarial data to update the model, we follow the same setting as [26], i.e., we set the perturbation bound to $\epsilon = 8/255$; PGD step number to $K = 10$; and PGD step size to $\alpha = 2/255$. We train the model by using SGD with momentum= 0.9 and learning rate $\eta = 0.01$. The number of communication rounds is set to $T = 150$ and the number of local epochs is set to $E = 1$. All methods use FedAvg for aggregation and use the same CNN network [23] on CIFAR10, CIFAR100, and SVHN datasets. We adopt Alexnet [14] to train the ImageNet subset for all methods. Recall that, compared with the cross-device setting, FAT matters more in the cross-silo setting, in which the number of clients is relatively *small*, and each client has powerful computation resources to handle the computation cost of AT [21]. Thus, we set the number of clients to $m = 5$ by default, and in each epoch, all clients are involved in the training. Experiments with more clients can be referred to Table 7 in Appendix D.7. The experiments are run on a server with Intel(R) Xeon(R) Gold 5218R CPU, 64GB RAM, and 8 Tesla V100 GPUs.

## D.2 Per-class Performance of Different Clients

Table 4 shows the per-class performance of different clients on CIFAR10 dataset. In FedGAIRAT, due to highly skewed label distribution, the prediction of each client is highly biased to the majority classes, leading to high performance on the majority classes and low performance (even 0% accuracy) on the minority classes. By contrast, in CalFAT, each client has higher performance on most classes. For example, on client 1, the accuracy of class 8 (96.56%) of FedGAIRAT is higher than CalFAT, due to that the prediction is highly biased to class 8 on client 1 for FedGAIRAT. By contrast, the accuracy of other (minority) classes on client 1 of FedGAIRAT is much lower than CalFAT. These results show that the calibrated cross-entropy loss can indeed improve the performance on minority classes, and further improve the overall performance of the model.

Table 4: Per-class natural accuracy and robust accuracy (against PGD-20 attack) of different clients on CIFAR10 dataset.

| | Class | | 0 | 1 | 2 | 3 | 4 | 5 | 6 | 7 | 8 | 9 | Average |
|---|---|---|---|---|---|---|---|---|---|---|---|---|---|
| Natural | client 1 | FedGAIRAT | 47.45 | 0.00 | 0.00 | 0.00 | 0.00 | 0.00 | 16.91 | 0.00 | 96.56 | 27.30 | 18.82 |
| | | CalFAT(ours) | 56.61 | 88.06 | 54.41 | 27.66 | 37.67 | 65.16 | 52.02 | 74.03 | 89.02 | 54.09 | 59.87 |
| | client 2 | FedGAIRAT | 0.00 | 93.24 | 0.00 | 80.97 | 0.00 | 0.00 | 74.49 | 0.00 | 0.00 | 0.00 | 24.87 |
| | | CalFAT(ours) | 71.55 | 86.34 | 58.46 | 62.88 | 16.75 | 29.28 | 78.75 | 47.76 | 75.44 | 42.22 | 56.94 |
| | client 3 | FedGAIRAT | 57.41 | 0.00 | 0.00 | 0.00 | 0.00 | 69.81 | 0.07 | 69.96 | 95.01 | 0.00 | 29.23 |
| | | CalFAT(ours) | 90.11 | 82.41 | 41.48 | 40.41 | 17.42 | 64.65 | 74.96 | 58.18 | 74.25 | 22.08 | 56.60 |
| | client 4 | FedGAIRAT | 4.23 | 0.61 | 0.00 | 0.00 | 0.06 | 2.54 | 0.00 | 56.13 | 0.00 | 99.72 | 16.33 |
| | | CalFAT(ours) | 59.76 | 76.50 | 46.15 | 28.80 | 28.23 | 63.09 | 79.07 | 77.62 | 79.06 | 46.88 | 58.52 |
| | client 5 | FedGAIRAT | 0.00 | 0.00 | 63.33 | 0.00 | 77.49 | 8.86 | 0.00 | 0.00 | 0.00 | 0.00 | 14.97 |
| | | CalFAT(ours) | 70.48 | 74.60 | 51.58 | 69.32 | 56.94 | 48.69 | 57.86 | 57.65 | 80.16 | 43.06 | 61.03 |
| Robust | client 1 | FedGAIRAT | 35.53 | 0.00 | 0.00 | 0.00 | 0.00 | 0.00 | 3.61 | 0.00 | 87.08 | 10.25 | 13.65 |
| | | CalFAT(ours) | 29.28 | 59.57 | 19.24 | 4.80 | 6.98 | 31.42 | 10.79 | 37.27 | 63.71 | 18.02 | 28.11 |
| | client 2 | FedGAIRAT | 0.00 | 71.03 | 0.06 | 21.54 | 0.04 | 0.00 | 82.64 | 0.00 | 0.00 | 0.00 | 17.53 |
| | | CalFAT(ours) | 35.61 | 72.05 | 15.45 | 22.06 | 5.24 | 7.89 | 55.68 | 23.54 | 39.47 | 6.17 | 28.32 |
| | client 3 | FedGAIRAT | 66.94 | 0.00 | 0.00 | 0.00 | 0.00 | 39.56 | 0.00 | 53.36 | 25.35 | 0.00 | 18.52 |
| | | CalFAT(ours) | 38.59 | 36.92 | 27.43 | 7.06 | 2.01 | 26.73 | 27.29 | 39.53 | 40.30 | 20.25 | 26.61 |
| | client 4 | FedGAIRAT | 6.02 | 0.00 | 0.00 | 0.00 | 0.00 | 0.12 | 0.00 | 42.34 | 0.00 | 93.73 | 14.22 |
| | | CalFAT(ours) | 17.37 | 11.05 | 9.17 | 1.17 | 3.35 | 47.13 | 55.97 | 27.78 | 56.53 | 51.64 | 28.12 |
| | client 5 | FedGAIRAT | 0.00 | 0.00 | 9.78 | 0.00 | 97.60 | 8.63 | 0.00 | 0.00 | 0.00 | 0.00 | 11.60 |
| | | CalFAT(ours) | 44.55 | 41.50 | 28.10 | 4.33 | 11.18 | 29.74 | 17.07 | 21.32 | 25.71 | 20.35 | 24.39 |

## D.3 Per-class Average Performance

Figure 5 shows the per-class average performance on SVHN dataset.

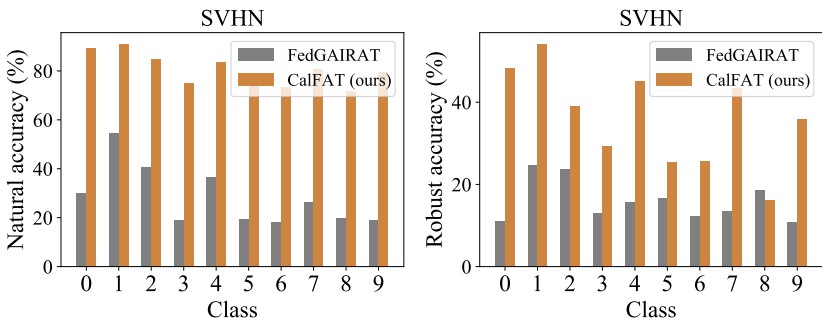

Figure 5: Per-class natural accuracy and robust accuracy (against PGD-20 attack) of CalFAT and the best baseline (FedGAIRAT) on SVHN dataset.

## D.4 Evaluation on Different Network Architectures

Table 5 shows the natural and robust accuracies with different network architectures on CIFAR10 dataset.

Table 5: Natural and robust accuracies (%) with different network architectures on CIFAR10 dataset.

| Network | CNN | | VGG-8 | | ResNet-18 | |
|---|---|---|---|---|---|---|
| Metric | Natural | PGD-20 | Natural | PGD-20 | Natural | PGD-20 |
| MixFAT | 53.23 | 26.22 | 59.60 | 34.99 | 67.54 | 38.25 |
| FedPGD | 47.21 | 26.50 | 62.21 | 34.89 | 65.48 | 30.04 |
| FedTRADES | 46.14 | 26.29 | 47.21 | 30.39 | 54.61 | 35.03 |
| FedMART | 25.68 | 18.15 | 43.28 | 30.16 | 52.13 | 33.24 |
| FedGAIRAT | 48.34 | 27.32 | 47.83 | 30.52 | 55.62 | 34.87 |
| FedRBN | 47.87 | 26.21 | 46.96 | 30.21 | 54.32 | 33.23 |
| CalFAT(ours) | **64.85** | **31.19** | **75.05** | **40.09** | **76.73** | **47.85** |

## D.5 Visualization of Different Methods

Figure 6 shows the t-SNE feature visualization of FedTRADES and CalFAT on SVHN dataset.

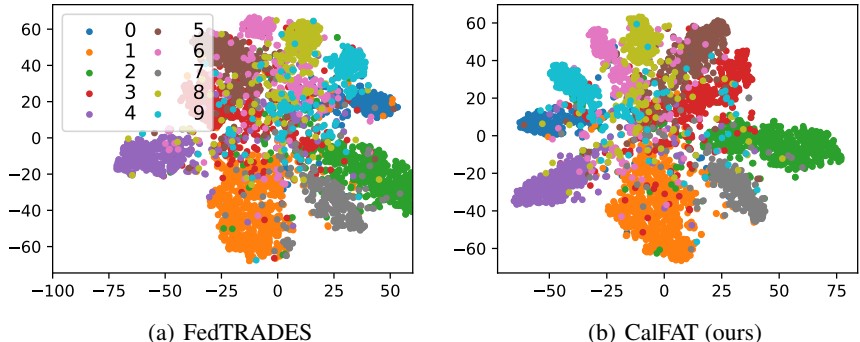

(a) FedTRADES          (b) CalFAT (ours)

Figure 6: t-SNE feature visualization of FedTRADES and CalFAT on SVHN dataset. Each color represents a class. Samples from different classes are hard to be separated in FedTRADES while CalFAT can learn more discriminative features.

## D.6 Performance under the IID setting

Table 6 shows the natural accuracy and robust accuracy (against PGD-20 attack) on CIFAR10 dataset under the IID setting.

Table 6: Natural and robust accuracy (%) on CIFAR10 dataset under the IID setting.

| Metric | Natural | PGD-20 |
|---|---|---|
| MixFAT | 79.62 | 37.57 |
| FedPGD | 75.89 | 42.16 |
| FedTRADES | 74.29 | 44.35 |
| CalFAT | 74.23 | 44.68 |

## D.7 Impact of the Number of Clients

Table 7 shows the natural and robust accuracies with different numbers of clients on CIFAR10 dataset.

Table 7: Natural and robust accuracies (%) with different numbers of clients $m = \{20, 50, 100\}$ on CIFAR10 dataset.

| $m$ | 20 | | | 50 | | | 100 | | |
|---|---|---|---|---|---|---|---|---|---|
| Metric | Natural | PGD-20 | AA | Natural | PGD-20 | AA | Natural | PGD-20 | AA |
| MixFAT | $26.59 \pm 0.16$ | $18.24 \pm 0.07$ | $13.12 \pm 0.14$ | $23.28 \pm 0.16$ | $15.55 \pm 0.13$ | $10.92 \pm 0.14$ | $20.85 \pm 0.16$ | $14.41 \pm 0.11$ | $10.66 \pm 0.12$ |
| FedPGD | $29.38 \pm 0.20$ | $18.19 \pm 0.18$ | $14.22 \pm 0.11$ | $27.73 \pm 0.15$ | $16.98 \pm 0.23$ | $11.94 \pm 0.14$ | $23.86 \pm 0.18$ | $15.37 \pm 0.18$ | $10.78 \pm 0.09$ |
| FedTRADES | $29.39 \pm 0.14$ | $18.47 \pm 0.13$ | $14.66 \pm 0.19$ | $21.44 \pm 0.06$ | $15.20 \pm 0.16$ | $11.85 \pm 0.09$ | $21.06 \pm 0.11$ | $14.76 \pm 0.16$ | $11.68 \pm 0.07$ |
| FedMART | $22.95 \pm 0.15$ | $17.08 \pm 0.07$ | $13.34 \pm 0.09$ | $22.43 \pm 0.15$ | $15.01 \pm 0.08$ | $11.59 \pm 0.06$ | $21.58 \pm 0.12$ | $14.48 \pm 0.17$ | $11.01 \pm 0.09$ |
| FedGAIRAT | $22.74 \pm 0.13$ | $17.00 \pm 0.12$ | $13.77 \pm 0.17$ | $20.84 \pm 0.26$ | $14.68 \pm 0.21$ | $11.80 \pm 0.17$ | $19.26 \pm 0.15$ | $14.17 \pm 0.11$ | $11.33 \pm 0.14$ |
| FedRBN | $21.90 \pm 0.13$ | $17.46 \pm 0.14$ | $12.91 \pm 0.11$ | $20.22 \pm 0.16$ | $14.74 \pm 0.16$ | $12.13 \pm 0.11$ | $18.99 \pm 0.11$ | $13.48 \pm 0.19$ | $12.05 \pm 0.08$ |
| CalFAT | $\mathbf{60.26 \pm 0.09}$ | $\mathbf{24.32 \pm 0.13}$ | $\mathbf{15.41 \pm 0.12}$ | $\mathbf{49.86 \pm 0.07}$ | $\mathbf{18.79 \pm 0.10}$ | $\mathbf{13.22 \pm 0.13}$ | $\mathbf{40.69 \pm 0.08}$ | $\mathbf{16.19 \pm 0.15}$ | $\mathbf{12.51 \pm 0.09}$ |

## D.8 Impact of Skewed Label Distribution

Table 8 shows the natural and robust accuracies under different level of label skewness on CIFAR10 dataset.

Table 8: Natural and robust accuracies (%) under different label skewness levels $\beta$ on CIFAR10 dataset.

| Label skewness level | $\beta = 0.05$ | | | | | | $\beta = 0.2$ | | | | | | $\beta = 0.3$ | | | | | |
|---|---|---|---|---|---|---|---|---|---|---|---|---|---|---|---|---|---|---|
| Metric | Natural | FGSM | BIM | CW | PGD-20 | AA | Natural | FGSM | BIM | CW | PGD-20 | AA | Natural | FGSM | BIM | CW | PGD-20 | AA |
| MixFAT | 49.10 | 27.49 | 25.32 | 22.17 | 25.24 | 22.51 | 54.85 | 31.27 | 28.70 | 26.08 | 28.46 | 25.21 | 58.93 | 31.68 | 28.17 | 24.96 | 28.00 | 24.34 |
| FedPGD | 47.13 | 26.63 | 24.96 | 20.75 | 25.03 | 21.28 | 52.22 | 30.31 | 28.64 | 25.49 | 28.59 | 24.92 | 56.12 | 30.86 | 28.46 | 25.07 | 28.29 | 23.64 |
| FedTRADES | 40.24 | 26.02 | 25.06 | 22.48 | 24.99 | 20.16 | 48.52 | 29.94 | 28.73 | 25.57 | 28.65 | 24.15 | 54.26 | 30.83 | 29.39 | 24.74 | 29.26 | 23.87 |
| FedMART | 29.84 | 21.90 | 21.39 | 18.31 | 21.41 | 17.89 | 38.38 | 27.59 | 27.05 | 23.31 | 26.99 | 21.89 | 40.96 | 28.32 | 27.88 | 23.12 | 27.80 | 22.16 |
| FedGAIRAT | 50.41 | 28.89 | 26.30 | 22.66 | 26.34 | 23.81 | 56.11 | 32.99 | 29.90 | 27.10 | 28.97 | 25.97 | 60.63 | 33.31 | 30.12 | 25.50 | 29.67 | 24.75 |
| FedRBN | 39.35 | 25.92 | 24.40 | 21.55 | 24.77 | 19.47 | 48.42 | 29.59 | 27.74 | 24.67 | 27.86 | 23.78 | 53.54 | 29.88 | 28.76 | 24.11 | 28.63 | 23.14 |
| CalFAT(ours) | **61.00** | **32.40** | **29.75** | **23.55** | **29.50** | **25.66** | **71.55** | **33.80** | **30.70** | **27.25** | **29.35** | **26.32** | **69.95** | **34.25** | **30.80** | **27.76** | **30.96** | **26.84** |

## D.9 Contribution of the Calibrated Loss Functions

Table 9 shows the results of different loss functions.

Table 9: Natural and robust accuracy (%) of different loss functions.

| Label skewness level | $\beta = 0.05$ | | | $\beta = 0.2$ | | | $\beta = 0.3$ | | |
|---|---|---|---|---|---|---|---|---|---|
| Metric | Natural | PGD-20 | AA | Natural | PGD-20 | AA | Natural | PGD-20 | AA |
| w/o $\ell_{cce}(\cdot, \cdot, \cdot)$ | 52.59 | 20.55 | 16.37 | 63.49 | 20.37 | 17.83 | 61.61 | 22.42 | 18.24 |
| w/o $\ell_{ckl}(\cdot, \cdot, \cdot)$ | 60.05 | 27.59 | 19.79 | 70.60 | 27.18 | 21.94 | 68.27 | 28.56 | 22.68 |
| CalFAT(ours) | **61.03** | **29.49** | **20.35** | **71.54** | **29.36** | **22.96** | **69.98** | **30.98** | **23.21** |

## D.10 Impact of the Ratio of Adversarial Data

Table 10 shows the robust accuracy (against PGD-20 attack) of CalFAT with different ratios of adversarial data.

Table 10: Robust accuracy (%) of our CalFAT against PGD-20 attack with different ratios of adversarial data.

| Ratio ($r$) | 0 | 0.3 | 0.5 | 0.8 | 1 |
|---|---|---|---|---|---|
| SVHN | 1.25 | 32.35 | 37.31 | 38.59 | **41.64** |
| CIFAR10 | 3.47 | 15.47 | 21.08 | 25.87 | **31.19** |
| CIFAR100 | 2.60 | 11.08 | 12.19 | 13.01 | **15.39** |