# OpenReview forum: "CalFAT: Calibrated Federated Adversarial Training with Label Skewness"
_NeurIPS.cc/2022/Conference — NeurIPS 2022 Accept_

### Official Review · Reviewer_RW5e · 2022-07-08

**Rating:** 5
**Confidence:** 4
**Soundness:** 2 fair
**Presentation:** 3 good
**Contribution:** 2 fair

**Summary:**

This paper proposes a calibrated federated adversarial training (CalFAT) method, which is used in the case of label skewness. Empirical evaluation is done on several datasets including SVHN, CIFAR-10/100 and a subset of ImageNet.

**Questions:**

Limited technical contribution and weak baselines.

**Strengths And Weaknesses:**

Strengths:
- The empirical evaluation is done on several different datasets, and under strong attacks like AA.
- The improvement in clean accuracy seems significant.
- The writing is clear and easy to follow the main idea.

Weaknesses:
- The technical contribution is quite limited. CalFAT is just a simple adaption of [22] into federated adversarial training.
- The compared baselines (e.g., MixFAT and Fed**) are off-the-shelf, which do not adapt to the special property of label skewness. Stronger baselines utilizing label skewness should be involved.
- The considered setting of federated learning with label skewness may be of limited interest to the community.

---

> ### Author Response · Authors · 2022-08-02
> **Response to Reviewer RW5e (Part 1)**
>
> Thanks for your valuable comments. We hope our response below can adequately address your concerns.
>
> ---
>
> **Q1:** The technical contribution is quite limited. CalFAT is just a simple adaption of [22] into federated adversarial training.
>
> **A1:**
> Thanks for the insightful question. While the logit calibration idea of our method follows [22] (termed LogitAdj), we would like to argue that we have made sufficient extension/improvement over LogitAdj:
>
> (1) We extend the logit calibration idea of [22] to the setting of FAT, and theoretically prove its usefulness in FAT via Propositions 1 and 2.
>
> (2) Technically, we use different loss functions from LogitAdj.
> Specifically, we use a calibrated KL loss for the local inner maximization and a calibrated CE loss for the local outer minimization, while LogitAdj only has a calibrated CE loss. As shown in Table 2 below, LogitAdj has much lower natural and robust accuracy than our CalFAT.
>
> (3) Please also note that the imbalance issue in centralized machine learning (majority class vs. minority class) is a different concept from the non-IID label skewness (different classes at different clients) in FAT. Such difference poses a notable difficulty in the theoretical analysis.
>
> We believe our exploration of logits calibration in FAT is of significant interest to the FL community. We hope the above clarifies the significance of our method.
>
> (Table 2: Combining FAT methods with different losses.)
> Metric | Natural | PGD-20 |
> |:------------------------------:|:-----------:|:-----------:|
> MixFAT | 53.35	$\pm$ 0.11 | 26.27	$\pm$ 0.11 |
> MixFAT + LogitAdj | 57.53	$\pm$ 0.21 | 27.65	$\pm$ 0.16 |
> MixFAT + RoBal | 58.25	$\pm$ 0.13 | 27.86	$\pm$ 0.10 |
> MixFAT + Calibration (ours) | **60.23**	$\pm$ 0.19 | **28.67**	$\pm$ 0.14 |
> ||
> FedPGD | 46.96	$\pm$ 0.16 | 26.74	$\pm$ 0.18 |
> FedPGD + LogitAdj | 59.79	$\pm$ 0.15 | 28.84	$\pm$ 0.12 |
> FedPGD + RoBal | 61.48	$\pm$ 0.07 | 29.51	$\pm$ 0.07 |
> FedPGD + Calibration (ours) | **63.91**	$\pm$ 0.13 | **30.72**	$\pm$ 0.16 |
> ||
> FedTRADES | 46.06	$\pm$ 0.12 | 26.31	$\pm$ 0.12 |
> FedTRADES + LogitAdj | 58.26	$\pm$ 0.20 | 27.92	$\pm$ 0.19 |
> FedTRADES + RoBal | 59.25	$\pm$ 0.23 | 28.63	$\pm$ 0.08 |
> FedTRADES + Calibration (ours) | **63.12**	$\pm$ 0.10 | **30.27**	$\pm$ 0.23 |
> ||
> FedMART | 25.67	$\pm$ 0.21 | 18.10	$\pm$ 0.22 |
> FedMART + LogitAdj | 42.01	$\pm$ 0.10 | 24.92	$\pm$ 0.02 |
> FedMART + RoBal | 44.26	$\pm$ 0.22 | 25.57	$\pm$ 0.17 |
> FedMART + Calibration (ours) | **48.85**	$\pm$ 0.08 | **27.19**	$\pm$ 0.11 |
> ||
> CalFAT (ours) | **64.69**	$\pm$ 0.08 | **31.12**	$\pm$ 0.11 |
>
> ---
>
> **Q2:** The compared baselines (e.g., MixFAT and Fed**) are off-the-shelf, which do not adapt to the special property of label skewness. Stronger baselines utilizing label skewness should be involved.
>
> **A2:**
> Thanks for the valuable suggestion.
> We choose MixFAT and FedRBN as baselines because they are the latest FAT methods.
> However, we have followed your suggestion and explored two additional losses that better utilize the label skewness property. More specifically, we adopted 2 long-tail learning losses (LogitAdj and RoBal) and combined them with FAT methods. The results are presented in Table 2 below where it shows that: 1) our calibration loss delivers better performance than both LogitAdj and RoBal; 2) CalFAT performs the best amongst all FAT baselines.
>
> (Table 2: Combining FAT methods with different losses.)
> |Metric | Natural | PGD-20 |
> |:------------------------------:|:-----------:|:-----------:|
> |MixFAT | 53.35	$\pm$ 0.11 | 26.27	$\pm$ 0.11 |
> |MixFAT + LogitAdj | 57.53	$\pm$ 0.21 | 27.65	$\pm$ 0.16 |
> |MixFAT + RoBal | 58.25	$\pm$ 0.13 | 27.86	$\pm$ 0.10 |
> |MixFAT + Calibration (ours) | **60.23**	$\pm$ 0.19 | **28.67**	$\pm$ 0.14 |
> ||
> |FedPGD | 46.96	$\pm$ 0.16 | 26.74	$\pm$ 0.18 |
> |FedPGD + LogitAdj | 59.79	$\pm$ 0.15 | 28.84	$\pm$ 0.12 |
> |FedPGD + RoBal | 61.48	$\pm$ 0.07 | 29.51	$\pm$ 0.07 |
> |FedPGD + Calibration (ours) | **63.91**	$\pm$ 0.13 | **30.72**	$\pm$ 0.16 |
> ||
> |FedTRADES | 46.06	$\pm$ 0.12 | 26.31	$\pm$ 0.12 |
> |FedTRADES + LogitAdj | 58.26	$\pm$ 0.20 | 27.92	$\pm$ 0.19 |
> |FedTRADES + RoBal | 59.25	$\pm$ 0.23 | 28.63	$\pm$ 0.08 |
> |FedTRADES + Calibration (ours) | **63.12**	$\pm$ 0.10 | **30.27**	$\pm$ 0.23 |
> ||
> |FedMART | 25.67	$\pm$ 0.21 | 18.10	$\pm$ 0.22 |
> |FedMART + LogitAdj | 42.01	$\pm$ 0.10 | 24.92	$\pm$ 0.02 |
> |FedMART + RoBal | 44.26	$\pm$ 0.22 | 25.57	$\pm$ 0.17 |
> |FedMART + Calibration (ours) | **48.85**	$\pm$ 0.08 | **27.19**	$\pm$ 0.11 |
> ||
> CalFAT (ours) | **64.69**	$\pm$ 0.08 | **31.12**	$\pm$ 0.11 |

---

> ### Author Response · Authors · 2022-08-02
> **Response to Reviewer RW5e (Part 2)**
>
> **Q3:** The considered setting of federated learning with label skewness may be of limited interest to the community.
>
> **A3:** Thanks for the thoughtful comment. We agree that there exist other types of non-IID settings, each poses unique challenges that require different techniques to tackle with [a]. The label skewness considered in this paper is arguably the most representative non-IID setting in the literature. There are an extensive body of works that specifically focus on only label skewness in federated learning, which manifests the importance of label skewness in federated learning, e.g. [b, c, d].
>
> It is arguably difficult to defend all non-IID settings via ne single method. We consider this as one limitation of our work as well as other works. We have clarified and discussed this point in the revision (Section 5, Line 364-366).
>
> We hope the above clarifies the significance of our method. We genuinely hope that, as the first work that reveals and properly tackles one major cause of the training instability and much degraded natural accuracy in FAT, our work is not penalized by not being able to cover all the non-IID settings.
>
> ---
>
> **References:**
>
> [a] Li, Xiaoxiao, et al. "FedBN: Federated Learning on Non-IID Features via Local Batch Normalization." International Conference on Learning Representations. 2020.
>
> [b] Li, Tian, et al. "Federated optimization in heterogeneous networks." Proceedings of Machine Learning and Systems 2 (2020): 429-450.
>
> [c] Shoham, Neta, et al. "Overcoming forgetting in federated learning on non-iid data." arXiv preprint arXiv:1910.07796 (2019).
>
> [d] T Dinh, Canh, Nguyen Tran, and Josh Nguyen. "Personalized federated learning with moreau envelopes." Advances in Neural Information Processing Systems 33 (2020): 21394-21405.

---

> ### Author Response · Authors · 2022-08-06
> **A follow up message**
>
> Dear Reviewer RW5e , Thanks again for the valuable comments.
>
> We have now clarified the significance and the limitations of our CalFAT and also show the new empirical results on Table 2. Note that more detailed information is shown in our rebuttal summary.
>
> Please kindly let us know if anything is unclear. We truly appreciate this opportunity to improve our work and shall be most grateful for any feedback you could give to us.

---

> ### Author Response · Authors · 2022-08-08
> **Gentle reminder for response**
>
> We would like to gently remind the reviewer of any follow-up clarifications or questions that we can do our best to address in the remaining limited time. We hope our previous response has clarified your comments on our technical contribution and limitations, and it has helped improved your opinion of our work. We truly appreciate all your comments and will incorporate them in our revised version. Please let us know if there are additional comments you have for us.

---

> ### Comment · Reviewer_RW5e · 2022-08-09
> **Thank you for the clarification**
>
> I thank the authors for further clarification and additional experiments, which solved most of my concerns. I would like to improve my score.

---

### Official Review · Reviewer_sKZG · 2022-07-08

**Rating:** 3
**Confidence:** 4
**Soundness:** 2 fair
**Presentation:** 2 fair
**Contribution:** 2 fair

**Summary:**

The paper discusses the cause of instability of federated adversarial learning(FAT) is the skewed label distribution. An adapted calibrated loss is applied to FAT to reduce the heterogeneity of local models and further improve the final model’s natural and robust accuracy. Experimental results of four computer vision datasets are conducted to support the author’s arguments.

**Questions:**

1 Line 14-16, I read that the authors claim they theoretically show a much improved convergence rate, expecting a theoretical proof on convergence rate. Prop 1 and Prop 2 in a combined manner show a better finall convergence point, as opposed to a faster convergence rate. I think these are two different things.


2 Line 259-260 says CKL loss can better generate adversarial examples. I suppose LogitAdj and RoBal do not associate with adversarial examples. Then my questions are:
    2.1) Do previous FAT baseline methods use the same adversarial training based on the same
           set(eg, same class ratio, sample size, etc) of adversarial examples?
    2.2) What’s the setup to apply LogitAdj/RoBal comparing with CalFAT?
    2.3) Is it possible to combine LogitAdj/RoBal with previous FAT baselines to compete with
           CalFAT?


3 Some experimental results are hard to tell the significance. For example, most numbers of AA column in Table 4 and Table 5 are very close. If the authors have conducted dfferent trials, standard deviation should be put there.


**Limitations:**

Yes

**Strengths And Weaknesses:**

Strength

1 The authors provide extensive experimental results.
2 The paper is well-written and easy to follow.

Weaknees

My biggest concern is the experiment design does not align well with the arguments.

The authors argue that the training instability comes from the skewed label distribution that leads to heterogeneity of the local model. I think it’d be true for both standard federated learning and federated adversarial training(FAT).  Eq.(11) is then proposed to calibrate the heterogeneity of local models for FAT. Therefore I expect at least two sets of experiments which I don’t see many of them in the main paper.

a) FAT baselines vs FAT baselines + Eq (11)
For example in Table 1, showing the results of {MixFAT, FedPGD, FedTRADES…} + Eq(11) would be more helpful to convince reviewer about the effectiveness of the proposed loss.

b) FAT baselines + other label distribution calibration vs FAT baselines + Eq(11)
Table 3 is a good example. However, the authors should explore it more. See Question 2 below.




Minor:
The structure of the paper can be further polished. For example, the table 1 and table 2 are actually the same experiments, and should be combined into one.

Line 241-242 is not necessary. I don’t see it helping validate any arguments made by authors.

---

> ### Author Response · Authors · 2022-08-02
> **Response to Reviewer sKZG (Part 1)**
>
> Thanks for your time reviewing our paper and the thoughtful comments. Following your suggestions, we have run additional experiments and added the new results to the revised paper. We hope the new version can adequately address your concerns. We are very happy to run more experiments if you have further concerns.
>
> ---
>
> **Q1:** More experiments: a) FAT baselines vs FAT baselines + Eq (11).
> b) FAT baselines + other label distribution calibration vs FAT baselines + Eq(11).
>
> **A1:**
> Thanks for your suggestions. Following your suggestion, we have combined FAT baselines with different losses (including LogitAdj, RoBal, and our calibration loss) for all the FAT baselines. The results indicate that our calibration loss-based methods achieve better performance than LogitAdj-based methods and RoBal-based methods; and our CalFAT performs the best amongst all the baselines. Please refer to the updated Table 2 in the revision for more details.
>
> We hope these new results have addressed your concerns about the comparison with other FAT baselines with different losses.
>
> (Table 2: Combining FAT methods with different losses.)
> Metric | Natural | PGD-20 |
> |:------------------------------:|:-----------:|:-----------:|
> |MixFAT | 53.35	$\pm$ 0.11 | 26.27	$\pm$ 0.11 |
> |MixFAT + LogitAdj | 57.53	$\pm$ 0.21 | 27.65	$\pm$ 0.16 |
> |MixFAT + RoBal | 58.25	$\pm$ 0.13 | 27.86	$\pm$ 0.10 |
> |MixFAT + Calibration (ours) | **60.23**	$\pm$ 0.19 | **28.67**	$\pm$ 0.14 |
> ||
> |FedPGD | 46.96	$\pm$ 0.16 | 26.74	$\pm$ 0.18 |
> |FedPGD + LogitAdj | 59.79	$\pm$ 0.15 | 28.84	$\pm$ 0.12 |
> |FedPGD + RoBal | 61.48	$\pm$ 0.07 | 29.51	$\pm$ 0.07 |
> |FedPGD + Calibration (ours) | **63.91**	$\pm$ 0.13 | **30.72**	$\pm$ 0.16 |
> ||
> |FedTRADES | 46.06	$\pm$ 0.12 | 26.31	$\pm$ 0.12 |
> |FedTRADES + LogitAdj | 58.26	$\pm$ 0.20 | 27.92	$\pm$ 0.19 |
> |FedTRADES + RoBal | 59.25	$\pm$ 0.23 | 28.63	$\pm$ 0.08 |
> |FedTRADES + Calibration (ours) | **63.12**	$\pm$ 0.10 | **30.27**	$\pm$ 0.23 |
> ||
> |FedMART | 25.67	$\pm$ 0.21 | 18.10	$\pm$ 0.22 |
> |FedMART + LogitAdj | 42.01	$\pm$ 0.10 | 24.92	$\pm$ 0.02 |
> |FedMART + RoBal | 44.26	$\pm$ 0.22 | 25.57	$\pm$ 0.17 |
> |FedMART + Calibration (ours) | **48.85**	$\pm$ 0.08 | **27.19**	$\pm$ 0.11 |
> ||
> |CalFAT (ours) | **64.69**	$\pm$ 0.08 | **31.12**	$\pm$ 0.11 |
>
> ---
>
> **Q2:** The structure of the paper can be further polished. For example, the table 1 and table 2 are actually the same experiments, and should be combined into one.
>
> **A2:** Thanks for the thoughtful suggestion. We have now combined them into one table, please find the new table (Table 1) in the revision.
>
> ---
>
> **Q3:** Line 241-242 is not necessary. I don’t see it helping validate any arguments made by authors.
>
> **A3:** Thanks for the suggestion. We have removed Line 241-242.

---

> ### Author Response · Authors · 2022-08-02
> **Response to Reviewer sKZG (Part 2)**
>
> **Q4:** Line 14-16, I read that the authors claim they theoretically show a much improved convergence rate, expecting a theoretical proof on convergence rate. Prop 1 and Prop 2 in a combined manner show a better final convergence point, as opposed to a faster convergence rate. I think these are two different things.
>
> **A4:** Thanks for pointing this out. We apologize for the confusion, by "much improved convergence rate" we mean "better convergence point" in this work. We have adjusted the wording at Line 14-16 in the revision to make it clear.
>
> ---
>
> **Q5:** Line 259-260 says CKL loss can better generate adversarial examples. I suppose LogitAdj and RoBal do not associate with adversarial examples. Then my questions are: 2.1) Do previous FAT baseline methods use the same adversarial training based on the same
> set(eg, same class ratio, sample size, etc) of adversarial examples? 2.2) What’s the setup to apply LogitAdj/RoBal comparing with CalFAT? 2.3) Is it possible to combine LogitAdj/RoBal with previous FAT baselines to compete with CalFAT?
>
> **A5:**
> Thanks for the insightful question.
>
> 2.1) Different FAT methods share the same hyperparameters (e.g., same class ratio, sample size, etc.) but use different methods to generate the adversarial examples. I.e., the only difference is the adversarial examples used to train their models.
>
> 2.2) \& 2.3)
> Thanks for your suggestions.
> We have combined different losses (including LogitAdj, RoBal, and our calibration loss) with previous FAT methods and report their performance in Table 2 below (as well as in the revised paper).
> Note that here all methods share the same settings except the loss function.
> Table 2 shows that: (1) our calibration loss-based methods achieved better performance than LogitAdj-based methods and RoBal-based methods; (2) CalFAT is the best among all FAT baselines.
> Please kindly refer to Table 2 in Section 4.1 in the revision for more details.
>
> We hope these results can help ease your concerns on the advantage of our CalFAT method.
>
> (Table 2: Combining FAT methods with different losses.)
> Metric | Natural | PGD-20 |
> |:------------------------------:|:-----------:|:-----------:|
> |MixFAT | 53.35	$\pm$ 0.11 | 26.27	$\pm$ 0.11 |
> |MixFAT + LogitAdj | 57.53	$\pm$ 0.21 | 27.65	$\pm$ 0.16 |
> |MixFAT + RoBal | 58.25	$\pm$ 0.13 | 27.86	$\pm$ 0.10 |
> |MixFAT + Calibration (ours) | **60.23**	$\pm$ 0.19 | **28.67**	$\pm$ 0.14 |
> ||
> |FedPGD | 46.96	$\pm$ 0.16 | 26.74	$\pm$ 0.18 |
> |FedPGD + LogitAdj | 59.79	$\pm$ 0.15 | 28.84	$\pm$ 0.12 |
> |FedPGD + RoBal | 61.48	$\pm$ 0.07 | 29.51	$\pm$ 0.07 |
> |FedPGD + Calibration (ours) | **63.91**	$\pm$ 0.13 | **30.72**	$\pm$ 0.16 |
> ||
> |FedTRADES | 46.06	$\pm$ 0.12 | 26.31	$\pm$ 0.12 |
> |FedTRADES + LogitAdj | 58.26	$\pm$ 0.20 | 27.92	$\pm$ 0.19 |
> |FedTRADES + RoBal | 59.25	$\pm$ 0.23 | 28.63	$\pm$ 0.08 |
> |FedTRADES + Calibration (ours) | **63.12**	$\pm$ 0.10 | **30.27**	$\pm$ 0.23 |
> ||
> |FedMART | 25.67	$\pm$ 0.21 | 18.10	$\pm$ 0.22 |
> |FedMART + LogitAdj | 42.01	$\pm$ 0.10 | 24.92	$\pm$ 0.02 |
> |FedMART + RoBal | 44.26	$\pm$ 0.22 | 25.57	$\pm$ 0.17 |
> |FedMART + Calibration (ours) | **48.85**	$\pm$ 0.08 | **27.19**	$\pm$ 0.11 |
> ||
> |CalFAT (ours) | **64.69**	$\pm$ 0.08 | **31.12**	$\pm$ 0.11 |

---

> ### Author Response · Authors · 2022-08-02
> **Response to Reviewer sKZG (Part 3)**
>
> **Q6:** Some experimental results are hard to tell the significance. For example, most numbers of AA column in Table 4 and Table 5 are very close. If the authors have conducted different trials, standard deviation should be put there.
>
> **A6:**
> Thanks for the suggestion. We have run all the experiments for 5 times and added the mean and standard deviation to Table 3 (original Table 4) in Section 4 and Table 7 (original Table 5) in Appendix D.7 in the revision.
> Since AA is a strong adversary, most methods can hardly improve the robustness under AA [5]. Nevertheless, our CalFAT can still improve the robustness of baseline FAT methods against AA as shown in Table 3 and Table 7 below. We hope these results can help address your concerns.
>
> (Table 3: Natural and robust accuracy (\%) across different FL frameworks on CIFAR10 dataset.)
> |FL framework |  FedProx |-|-| Scaffold |-|-|
> |:------------:|:-----------:|:-----------:|:-----------:|:-----------:|:-----------:|:-----------:|
> |Metric | Natural | PGD-20 | AA | Natural | PGD-20 | AA |
> ||
> |MixFAT | 53.75	$\pm$ 0.16 | 29.61	$\pm$ 0.19 | 21.59	$\pm$ 0.27 | 55.27	$\pm$ 0.20 | 28.78	$\pm$ 0.15 | 21.26	$\pm$ 0.11 |
> |FedPGD | 49.57	$\pm$ 0.18 | 28.48	$\pm$ 0.17 | 21.31	$\pm$ 0.18 | 49.52	$\pm$ 0.14 | 27.46	$\pm$ 0.21 | 20.27	$\pm$ 0.15 |
> |FedTRADES | 48.14	$\pm$ 0.20 | 27.75	$\pm$ 0.17 | 21.13	$\pm$ 0.21 | 47.78	$\pm$ 0.23 | 27.31	$\pm$ 0.16 | 20.04	$\pm$ 0.16 |
> |FedMART | 28.32	$\pm$ 0.22 | 19.32	$\pm$ 0.23 | 15.91	$\pm$ 0.25 | 27.80	$\pm$ 0.17 | 20.03	$\pm$ 0.26 | 16.85	$\pm$ 0.15 |
> |FedGAIRAT | 49.61	$\pm$ 0.20 | 29.34	$\pm$ 0.11 | 21.33	$\pm$ 0.18 | 49.54	$\pm$ 0.21 | 27.23	$\pm$ 0.25 | 20.16	$\pm$ 0.09 |
> |FedRBN | 47.26	$\pm$ 0.13 | 26.63	$\pm$ 0.15 | 20.46	$\pm$ 0.06 | 49.77	$\pm$ 0.09 | 28.37	$\pm$ 0.12 | 20.32	$\pm$ 0.06 |
> |CalFAT | **66.32**	$\pm$ 0.08 | **32.79**	$\pm$ 0.13 | **22.83**	$\pm$ 0.11 | **67.16**	$\pm$ 0.06 | **32.94**	$\pm$ 0.06 | **21.94**	$\pm$ 0.05 |
>
> (Table 7: Natural and robust accuracy (\%) across different numbers of clients $m=\{20,50,100\}$ on CIFAR10 dataset.)
> |$m$ | 20 | - | - | 50 | - | -  | 100 | - | -  |
> |:---------:|:-----------:|:-----------:|:-----------:|:-----------:|:-----------:|:-----------:|:-----------:|:-----------:|:-----------:|
> |Metric | Natural | PGD-20 | AA | Natural | PGD-20 | AA | Natural | PGD-20 | AA |
> ||
> |MixFAT | 26.59	$\pm$ 0.16 | 18.24	$\pm$ 0.07 | 13.12	$\pm$ 0.14 | 23.28	$\pm$ 0.16 | 15.55	$\pm$ 0.13 | 10.92	$\pm$ 0.14 | 20.85	$\pm$ 0.16 | 14.41	$\pm$ 0.11 | 10.66	$\pm$ 0.12 |
> |FedPGD | 29.38	$\pm$ 0.20 | 18.19	$\pm$ 0.18 | 14.22	$\pm$ 0.11 | 27.73	$\pm$ 0.15 | 16.98	$\pm$ 0.23 | 11.94	$\pm$ 0.14 | 23.86	$\pm$ 0.18 | 15.37	$\pm$ 0.18 | 10.78	$\pm$ 0.09 |
> |FedTRADES | 29.39	$\pm$ 0.14 | 18.47	$\pm$ 0.13 | 14.66	$\pm$ 0.19 | 21.44	$\pm$ 0.06 | 15.20	$\pm$ 0.16 | 11.85	$\pm$ 0.09 | 21.06	$\pm$ 0.11 | 14.76	$\pm$ 0.16 | 11.68	$\pm$ 0.07 |
> |FedMART | 22.95	$\pm$ 0.15 | 17.08	$\pm$ 0.07 | 13.34	$\pm$ 0.09 | 22.43	$\pm$ 0.15 | 15.01	$\pm$ 0.08 | 11.59	$\pm$ 0.06 | 21.58	$\pm$ 0.12 | 14.48	$\pm$ 0.17 | 11.01	$\pm$ 0.09 |
> |FedGAIRAT | 22.74	$\pm$ 0.13 | 17.00	$\pm$ 0.12 | 13.77	$\pm$ 0.17 | 20.84	$\pm$ 0.26 | 14.68	$\pm$ 0.21 | 11.80	$\pm$ 0.17 | 19.26	$\pm$ 0.15 | 14.17	$\pm$ 0.11 | 11.33	$\pm$ 0.14 |
> |FedRBN | 21.90	$\pm$ 0.13 | 17.46	$\pm$ 0.14 | 12.91	$\pm$ 0.11 | 20.22	$\pm$ 0.16 | 14.74	$\pm$ 0.16 | 12.13	$\pm$ 0.11 | 18.99	$\pm$ 0.11 | 13.48	$\pm$ 0.19 | 12.05	$\pm$ 0.08 |
> |CalFAT | **60.26**	$\pm$ 0.09 | **24.32**	$\pm$ 0.13 | **15.41**	$\pm$ 0.12 | **49.86**	$\pm$ 0.07 | **18.79**	$\pm$ 0.10 | **13.22**	$\pm$ 0.13 | **40.69**	$\pm$ 0.08 | **16.19**	$\pm$ 0.15 | **12.51**	$\pm$ 0.09 |

---

> ### Author Response · Authors · 2022-08-06
> **Thanks to Reviewer sKZG**
>
> We would like to thank the reviewer for taking the time to review our paper and for the valuable comments.
>
> Kindly let us know whether we have adequately addressed your comments on the experiment design and writing. Note that more detailed information is provided in our rebuttal summary.
>
> We truly appreciate this opportunity to improve our work and shall be most grateful for any feedback you could give to us.

---

> ### Author Response · Authors · 2022-08-08
> **Gentle reminder for response**
>
> We would like to gently remind the reviewer of any follow-up clarifications or questions that we can do our best to address in the remaining time. We hope our previous response has clarified your comments and it has helped improved your opinion of our work. We truly appreciate all your comments and will incorporate them in our revised version. Please let us know if there are additional comments you have for us.

---

### Official Review · Reviewer_tX9B · 2022-07-10

**Rating:** 8
**Confidence:** 4
**Soundness:** 4 excellent
**Presentation:** 4 excellent
**Contribution:** 4 excellent

**Summary:**

This paper aims to answer the question of how to guarantee adversarial robustness in federated learning. In particular, this paper targets a challenging non-IID setting - the skewed label distribution, which gives rise to imbalanced class probabilities and heterogeneous local models, and proposed CalFAT to solve this problem by adaptively adjusting the logits. Theoretically, the authors prove that CalFAT can help learn homogeneous local models. Empirically, CalFAT outperforms other traditional centralized adversarial training methods (like AT, TRADES, MART, GAIRAT) and also the recent robustness propagation method (I.e., FedRBN) on several benchmarked datasets (like CIFAR-10, SVHN, CIFAR100).

**Questions:**

- Why the natural performance degradation is considered as the main motivation (Figure 1 on Page 2) instead of robust performance since the goal of FAT is gaining robustness?
- I am wondering whether CalFAT also works in iid setting, can you elaborate more on this?



**Ethics Review Area:**

["I don’t know"]

**Limitations:**

The authors did not discuss the limitations of the work.

**Strengths And Weaknesses:**

Strengths:
+ The literature review is solid; summarizing the advantages and disadvantages of the literature are very helpful.
+ The paper points out why the previous methods hurt convergence of FAT and proposes novel techniques to tackle this issue that leads to better convergence and higher performance.
+ The paper provides a solid theoretical analysis.
+ Experimental evaluations are solid and well designed. The comparisons include the most recent FAT and centralized AT methods.
+ The results are convincing and show that CalFAT outperforms state-of-the-arts.

Weaknesses:
- It is not clear why the natural performance degradation is considered as the main motivation (figure 1 on Page 2) instead of robust performance since the goal of FAT is gaining robustness. Adding more elaboration would be helpful.
- More discussions on the potential applications of FAT are expected to better highlight its significance.
- Even though sufficient references are given, a few relevant papers are recommended: “Privacy and Robustness in Federated Learning: Attacks and Defenses” that also touched on federated adversarial robustness; “Decision Boundary-aware Data Augmentation for Adversarial Training” that studied how to improve adversarial robustness by playing with decision boundary. These papers did not study exactly the same topic as this paper, but would certainly enrich the literature review further.

---

> ### Author Response · Authors · 2022-08-02
> **Response to Reviewer tX9B**
>
> Thank you very much for the positive feedback and valuable comments. We hope the following new results and clarifications can address your concerns.
>
> ---
>
> **Q1:** It is not clear why the natural performance degradation is considered as the main motivation (figure 1 on Page 2) instead of robust performance since the goal of FAT is gaining robustness. Adding more elaboration would be helpful.
>
> **A1:** Thanks for the insightful question. While FAT improves adversarial robustness, it also significantly reduces the natural accuracy [26], making the final model less usable in real-world scenarios. Therefore, we aim to address this low natural accuracy issue of FAT without affecting the robust performance.
> In other words, robustness is also one key part of our goal but was tackled through the improvement of natural accuracy and stable training. As indicated by the results in Figure 1, the training instability issue and low natural accuracy appear to be the major cause of degraded robustness.
>
> We hope the above clarifies your concern.
>
> ---
>
> **Q2:** More discussions on the potential applications of FAT are expected to better highlight its significance.
>
> **A2:** Thanks for the suggestion. In fact, FAT can be applied in many real-world FL systems where robustness is one of the top concerns, e.g., disease diagnosis in medical applications or anti-money laundering in financial applications. In these scenarios, data is often non-IID, where our CalFAT can help train more robust models. We have added more discussions in the revision.
> Please refer to Section 1, Line 28-29.
>
> ---
>
> **Q3:** Even though sufficient references are given, a few relevant papers are recommended: “Privacy and Robustness in Federated Learning: Attacks and Defenses” that also touched on federated adversarial robustness; “Decision Boundary-aware Data Augmentation for Adversarial Training” that studied how to improve adversarial robustness by playing with decision boundary. These papers did not study exactly the same topic as this paper, but would certainly enrich the literature review further.
>
> **A3:** Thanks for the suggestion. We have cited and discussed these papers in the revision.
> Please refer to Section 1, Line 21-25 and 32-35.
>
> ---
>
> **Q4:** I am wondering whether CalFAT also works in iid setting, can you elaborate more on this?
>
> **A4:**
> Thanks for your valuable comments. We have conducted the experiments under iid setting.
> The results are reported in Table 6 below. It shows that our CalFAT achieves the best robustness (under PGD-20 attack).
> Please refer to Section 4.5 and Appendix D.6 in our revised version for details.
> We hope these results can help clarify your concern.
>
> (Table 6: Natural and robust accuracy (\%) on CIFAR10 dataset under the IID setting.)
> | Metric    | Natural | PGD-20 |
> |:-----------:|:---------:|:--------:|
> | MixFAT    | 79.62   | 37.57  |
> | FedPGD    | 75.89   | 42.16  |
> | FedTRADES | 74.29   | 44.35  |
> | CalFAT    | 74.23   | 44.68  |

---

> ### Author Response · Authors · 2022-08-06
> **Any additional questions?**
>
> Comment:
> We would like to thank the reviewer for the detailed comments, and particularly, for admitting our work with an interesting topic and good motivation.
>
> We hope our response has adequately addressed your concerns regarding the significance of FAT and related references. The experiment results in the iid setting are shown in Table 6. Note that more information can be found in our rebuttal summary.
>
> Kindly let us know if anything is unclear. We truly appreciate your valuable feedback and comments that help us further improve our work.

---

> > ### Comment · Reviewer_tX9B · 2022-08-08
> > **Thank the authors for the response.**
> >
> > I want to thank the authors for your response, which addressed my major concerns. I have improved my rating.

---

### Author Response · Authors · 2022-08-03
**Rebuttal Summary**

We sincerely thank all reviewers for their valuable comments and suggestions. We have made the following updates (where * denotes the content included in our initial submitted paper, and + denotes the newly added content during the rebuttal):

* *Abstract: change "much improved convergence rate" to "better convergence point".

+ +Section 1: added discussions about application of FAT; added references of relevant papers.

* *Section 4.1: reorganized the main experimental results of our CalFAT against other FAT methods under the non-IID setting.

+ +Section 4.1: added the new experimental results of our CalFAT against state-of-the-art long-tail learning methods under the non-IID setting; added standard deviation to the empirical results.

+ +Section 4.5: added the new experimental results of our CalFAT against other FAT methods under the IID setting.

+ +Section 5: added discussion about other non-IID settings in FAT.

+ +Appendix D.6: added the new experimental results of our CalFAT against other FAT methods under the IID setting.

* *Appendix D.7: the experimental results of our CalFAT against other FAT methods across different numbers of clients.

+ +Appendix D.7: added standard deviation to the empirical results.

* *Appendix D.8: the experimental results of our CalFAT against other FAT methods under different levels of label skewness.

We have revised our paper according to all the valuable comments and please let us know if there is anything still not clear. We are happy to run more experiments if you have further suggestions.

---

### Meta-Review · Area_Chair_ptk7 · 2022-08-29

**Recommendation:** Accept
**Confidence:** Certain

**Metareview:**

This submission aims to ensure adversarial robustness in federated learning when label skewness exists among different local agents. The main idea of the proposed solution is to calibrate the logits to balance the predicted marginal label probabilities. Most of the reivewers found the topic studied in this work relevant, important and timely. The authors have also successfully addressed the concerns from the reviewers during the rebuttal. Hence I recommend acceptance.

**Award:**

No

---

### Decision · Program_Chairs · 2022-09-14

Accept